# Experience is the Best Teacher:
# Motivating Effective Exploration in Reinforcement Learning for LLMs

**Wenjian Zhang** [1][2]  **Kongcheng Zhang** [3]  **Jiaxin Qi** [1][2]  **Jianqiang Huang** [1][2]  **Baisheng Lai** [1][2]

## Abstract

Reinforcement Learning (RL) with rubric-based rewards has recently shown remarkable progress in enhancing general reasoning capabilities of Large Language Models (LLMs), yet still suffers from ineffective exploration confined to current policy distribution. In fact, RL optimization can be viewed as steering the policy toward an ideal distribution that maximizes the rewards, while effective exploration should align efforts with desired target. Leveraging this insight, we propose HeRL, a **_Hindsight experience guided Reinforcement Learning_** framework to bootstrap effective exploration by explicitly *telling LLMs the desired behaviors* specified in rewards. Concretely, HeRL treats failed trajectories along with their unmet rubrics as hindsight experience, which serves as in-context guidance for the policy to explore desired responses beyond its current distribution. Additionally, we introduce a bonus reward to incentivize responses with greater potential for improvement under such guidance. HeRL facilitates effective learning from desired high-quality samples without repeated trial-and-error from scratch, yielding a more accurate estimation of the expected gradient theoretically. Extensive experiments across various benchmarks demonstrate that HeRL achieves superior performance gains over baselines, and can further benefit from experience guided self-improvement at test time. Our code is available at https://github.com/sikelifei/HeRL.

[1]Computer Network Information Center, Chinese Academy of Sciences, Beijing [2]University of Chinese Academy of Sciences, Beijing [3]Zhejiang University. Correspondence to: Jianqiang Huang <jqhuang@cnic.cn>, Baisheng Lai <bslai@cnic.cn>.

*Proceedings of the $43^{rd}$ International Conference on Machine Learning*, Seoul, South Korea. PMLR 306, 2026. Copyright 2026 by the author(s).

## 1. Introduction

> *"All of what we mean by goals and purposes can be well thought of as maximization of the expected value of the cumulative sum of a received scalar signal (reward)."* — Richard Sutton

Large Language Models (LLMs) have demonstrated impressive potential over a wide range of complex reasoning tasks, including mathematical analysis (Hendrycks et al., 2021; He et al., 2024; Zhang et al., 2025a), code generation (Chen et al., 2021; Wei et al., 2026), and robotic control (Driess et al., 2023; Huang et al., 2025a). One of the pivotal techniques driving these advancements is Reinforcement Learning with Verifiable Rewards (RLVR), where models optimize their reasoning through rule-based correctness verification (Guo et al., 2025a; OpenAI, 2025; Zeng et al., 2025). While effective in tasks with clear verifiable outcomes, the RLVR paradigm remains challenging in open-ended scenarios like healthcare (Qiu et al., 2024; Li et al., 2024; Arora et al., 2025) and instruction following (Zhou et al., 2023; He et al., 2025; Wen et al., 2024), as straightforward ground-truth labels are often unavailable.

To address this limitation, previous works typically employ reward models to assign scores that quantify how well the responses align with human judgments (Ouyang et al., 2022; Zhong et al., 2025). However, they inevitably need a large scale of human-annotated preference data for training and are prone to overfitting superficial patterns in the data, which may introduce bias and limit broader generalization (Yu et al., 2025a;b). More recently, rubric based reinforcement learning emerges as a promising way for general reasoning tasks, where checklist-style criteria are defined to assess multi-dimensional aspects of the response quality (Gunjal et al., 2025; Huang et al., 2025b; He et al., 2025). By leveraging "LLM-as-a-Judge" (Li et al., 2025) to score each criterion and combining feedback into scalar rewards, rubrics provide a scalable and interpretable alternative that extends the standard RLVR to open-ended scenarios.

Despite the encouraging results, existing methods still struggle to effectively explore high-quality samples for learning even after numerous rollouts, particularly for hard problems. Recent attempts like structured search (Hou et al.,

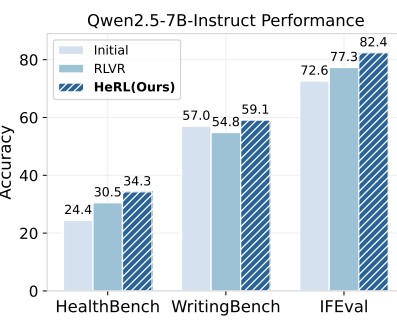
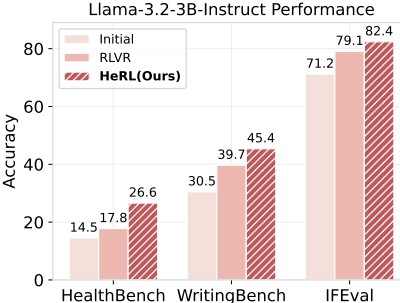

Figure 1. (Left) A conceptual illustration of the efficiency of exploration and the effectiveness of experience guided sampling. (Right) Model performance comparison between baselines and our proposed HeRL across different reasoning domains.

2025; Zheng et al., 2025) and intrinsic reward driven exploration (Yao et al., 2026; Dai et al., 2025) can produce more diverse responses than stochastic sampling, but they lack a principled consideration of *which exploration truly benefits optimization*. These approaches rely on blind trial-and-error from inherent policy distribution without knowing what constitutes desirable behaviors, potentially hindering further performance gains by failing to discover critical patterns. Under *reward hypothesis* (Sutton, 2004; Sutton & Barto, 2018), RL optimization can be viewed as steering the policy toward an ideal distribution that maximizes the expected rewards. This naturally leads to an intuitive question: **Can we directly bias the exploration toward high-reward responses for more efficient learning?** Fortunately, rubrics offer a promising path forward: beyond the aggregated scalar reward to indicate correctness, the model can easily obtain lesson-like experience guidance on how to discover better solutions in hindsight leveraging the language description of desired criteria (*e.g.*, style, coherence). This helps the model to align exploration with optimization target, thereby facilitating more effective learning.

Building on this insight, we propose HeRL, a __H__*indsight* __e__*xperience guided* __R__*einforcement* __L__*earning* framework to bootstrap effective exploration by explicitly *telling LLMs the desired behaviors* specified in rewards. Technically, HeRL treats initial failures along with their unmet rubrics as hindsight experience, which serves as in-context guidance for the policy to explore desired improvements beyond its current distribution. This guided exploration utilizes the rich language priors from rubrics, enabling the model to efficiently generate high-quality training samples without repeated trial-and-error from scratch. To further stabilize the exploration and exploitation of desired improvements, we introduce a bonus reward that prioritizes responses exhibiting greater improvement potential, coupled with a policy shaping mechanism that encourages learning from unfamiliar yet desirable improvements. HeRL performs RL training on both the initial responses and their subsequent improvements, theoretically leading to a more accurate estimation of the expected gradient in terms of optimizing objective.

Our core contributions are summarized as follows:

- We provide an in-depth analysis of *what makes effective exploration* in RL for LLM reasoning, highlighting that language priors from rewards can help to align exploration with desired outcomes for efficient learning. (§ 2)

- We propose HeRL, a novel RL framework that integrates hindsight experience into task instructions for guiding the discovery of desired high-quality responses, complemented by a bonus reward and policy-shaping mechanism to stabilize the exploration and exploitation. (§ 3)

- Extensive experiments on diverse open-ended tasks demonstrate that the proposed HeRL achieves results superior to existing counterparts. Notably, HeRL can further benefit from experience-guided self-improvement during test time, yielding continued performance gains. (§ 4)

## 2. Effective Exploration in RL for LLMs

In this section, we will clarify our insight: *What makes effective exploration in RL?* This stems from the classical *reward hypothesis* (Sutton, 2004; Sutton & Barto, 2018), where the ideal policy distribution converges to a high-reward region during training. This inspires us that effective exploration should align efforts with the optimization target. We will model and demonstrate this intuition in Section 2.1 by analyzing the learning mechanisms of explored samples. After gaining insight into how different samples contribute to optimization, we will empirically investigate the efficiency of various exploration strategies in Section 2.2.

### 2.1. Theoretical Intuition

The left side of Figure 1 illustrates the potential exploration directions in the solution space under different sampling strategies. To better understand the learning mechanisms under different explored samples, we analyze their token-level gradients. The gradient of the RL objective is (we omit

`min` and `clip` operations under on-policy settings):

$$\nabla \mathcal{J}(\theta) = \mathbb{E}_{q \sim \mathcal{D}, \, y \sim \pi_{\theta_{\text{old}}}(\cdot | q)} \left[ \sum_{t=1}^{|y|} A_t \nabla_\theta \log \pi_\theta(y_t | q, y_{<t}) \right],$$

(1)

where $\pi_\theta$ denotes the policy model, $y$ denotes response and $A_t$ denotes the advantage of the $t$-th token. To analyze how these gradients affect the model's token distribution, we further examine its propagation through the logits. Let $z_v = [z_1, z_2, ..., z_{|\mathcal{V}|}]$ denote the logit corresponding to token $v$ in vocabulary $\mathcal{V}$, let $\pi_\theta(v | q, y_{<t})$ denote the probability of token $v$ calculated by softmax, *i.e.*, $\pi_\theta(v | q, y_{<t}) = \exp(z_v) / \sum_{v'} \exp(z_{v'})$. Then we have the following gradient descent direction:

$$\frac{\partial \log \pi_\theta(y_t | q, y_{<t}) \cdot A_t}{\partial z_v} =$$
$$\begin{cases} \left(1 - \pi_\theta(y_t | q, y_{<t})\right) \cdot A_t & \text{if } v = y_t \text{ (sampled)} \\ -\pi_\theta(v | q, y_{<t}) \cdot A_t & \text{if } v \neq y_t \text{ (unsampled)} \end{cases}.$$

(2)

This formulation clearly shows how different explored samples work: high-quality samples with positive advantages increase the logit of sampled tokens and decrease the logits of all unexplored tokens, whereas low-quality samples with negative advantages induce the opposite effect, raising the logits of many unexplored tokens. In the reinforced fine-tuning of LLMs, the action space can be extremely large, while only a very small subset of actions is desirable under a given state. Consequently, when exploration fails to discover sufficient high-quality samples, the dominant negative gradients will diffuse to numerous irrelevant tokens, deviating from the intended distribution. This highlights the importance of aligning exploration with desired optimization target, which leads to stable and reliable updates. The derivation of Eq. (2) can be found in Appendix D.

## 2.2. Empirical Analysis

The analysis in Section 2.1 provides an intuition for prioritizing high-quality responses during exploration. However, the comparative efficiency among different sampling strategies remain unclear. Therefore, we investigate this empirically in the following.

We randomly select 500 questions from HealthBench (Arora et al., 2025) and evaluate its PassRate of different sampling stategies under the same number of attempts. Specifically, we compare three different sampling methods on Qwen2.5-7B-Instruct: (1) *stochastic sampling* with random seeds, (2) *entropy-based search* branching at high-entropy tokens, and (3) *guided sampling* by hindsight experience, which feeds the original response together with the unmet rubrics back to the model.

As shown in Figure 2, we find that guided sampling by

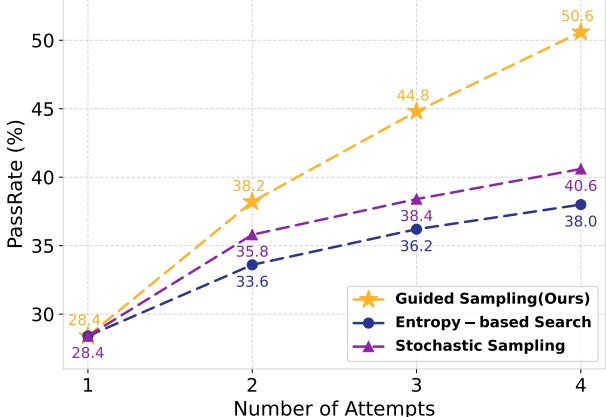

*Figure 2.* Performance comparison of sampling strategies. The guided sampling by hindsight experience consistently outperforms stochastic sampling and entropy-based sampling.

hindsight experience consistently outperforms stochastic sampling and entropy-based search under the same number of attempts. Notably, entropy-based search yields an even lower PassRate than stochastic sampling, as the large vocabulary space of LLMs makes token-level branching inefficient for search with limited sampling budgets. This demonstrates that guided sampling is capable of producing more desirable samples under the same inference cost, which provides strong evidence that using hindsight experience for exploration may effectively enhance RL performance by learning from desired high-quality responses.

## 3. Methodology

In this section, we introduce our HeRL framework, as illustrated in Figure 3. For each input instruction, we first sample a set of candidate trajectories from the current policy and evaluate them with verifiable rubrics to obtain both rewards and language feedback of unmet rubrics. We then treat this feedback and the original trajectory as hindsight experience to guide the policy to generate improvements that revise the unsatisfied rubrics, which shifts learning from pure scalar rewards to learning from explicit experience guidance. Finally, we use the successfully revised trajectories along with original trajectories for reinforcement learning training.

### 3.1. Hindsight Experience guided Exploration

Although revising all failed trajectories into successful ones is possible, we found it more effective to focus on the trajectories that are more likely to benefit from revision, *i.e.*, failures with the highest rewards. This is motivated by the Zone of Proximal Development (ZPD) (Ness, 2023), which suggests learning is more effective when feedback targets the gap between what the model can accomplish independently and what is achievable with guidance. Low-reward

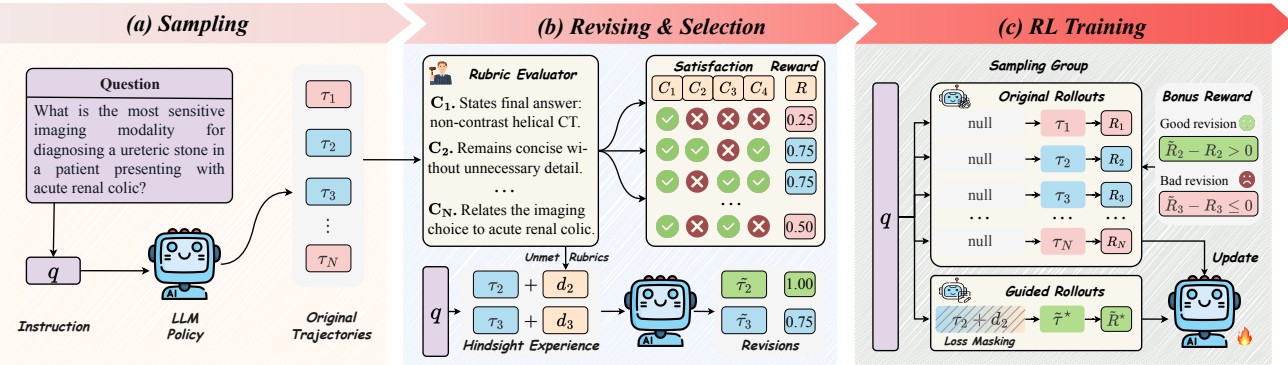

*Figure 3.* The overall framework of HeRL. First, we sample candidate trajectories and evaluate them using checklist-style rubrics. Then we revise failed trajectories with highest reward and preserve the best improvements. Both the original attempts and subsequent improvements are optimized using reinforcement learning, supplemented by a bonus reward to incentive responses with higher improvement potential.

trajectories are typically far from the desired behaviors, whereas high-reward trajectories lie closer to feasible solutions. Therefore, revising high-reward trajectories provides a more computationally efficient pathway for the model to break through its existing performance bottlenecks.

Given an instruction $q$, we first sample a group of $N$ trajectories from the current policy $\pi_\theta$:

$$\tau_i \sim \pi_\theta(\cdot \mid q), \quad i = 1, \ldots, N. \quad (3)$$

We then evaluate each trajectory $\tau_i$ to obtain the scalar reward $r_i = R(\tau_i; q)$ and language feedback of unmet rubrics $d_i$, where $R(\tau_i; q)$ is computed by *explicit aggregation* (Gunjal et al., 2025) over checklist-style rubrics. Specifically, for a rubric set $\mathcal{C} = \{c_j\}_{j=1}^K$, we employ a binary indicator $\mathbb{I}(q, \tau_i, c_j) \in \{0, 1\}$, which equals 1 if $\tau_i$ satisfies the rubric item $c_j$ and 0 otherwise. The scores of all rubric items are aggregate into a normalized weighted reward:

$$R(\tau_i; q) = \frac{\sum_{j=1}^K w_j \, \mathbb{I}(q, \tau_i, c_j)}{\sum_{j=1}^K w_j}, \quad (4)$$

where $w_j$ denotes the weight of the $j$-th rubric.

After that, we select a set of failed trajectories with the maximum rewards within the group for subsequent revision:

$$\mathcal{I}_{\text{top}} = \arg\max_i \left( \{r_i\}_{i=1}^N \right). \quad (5)$$

We prompt the policy to revise each selected trajectory conditioned on the hindsight experience $\mathcal{H}_i = (\tau_i, d_i)$, which serves as in-context guidance describing how the original trajectory can be improved. This hindsight experience provides reward priors that help to preserve satisfied rubrics while refining unmet ones. The policy model then generates a revised response $\tilde{\tau}_i$ and receives a new reward $\tilde{r}_i$:

$$\tilde{\tau}_i \sim \pi_\theta(\cdot \mid q, \mathcal{H}_i), \quad \tilde{r}_i = R(\tilde{\tau}_i; q), \quad i \in \mathcal{I}_{\text{top}}. \quad (6)$$

Among all revised candidates, we add one that achieves the maximum reward into the sampling group:

$$i^\star = \arg\max_{i \in \mathcal{I}_{\text{top}}} \tilde{r}_i, \quad \tilde{\tau}^\star \triangleq \tilde{\tau}_{i^\star}. \quad (7)$$

In addition, we introduce a bonus reward to encourage trajectories with higher improvement potential. The underlying principle is that a failed attempt remains valuable if it serves as a high-quality stepping stone for future success. Accordingly, we reward exploration not only for its current correctness, but also for its expected contribution to future performance gains. For each selected trajectory $i \in \mathcal{I}_{\text{top}}$, after generating a revised response $\tilde{\tau}_i$ and receiving its reward $\tilde{r}_i$, we augment the original reward $r_i$ with a bonus that reflects the potential for improvement under guidance:

$$r_i \leftarrow r_i + \alpha\big(\tilde{r}_i - r_i\big), \quad (8)$$

where $i \in \mathcal{I}_{\text{top}}$, and $\alpha$ is set to 0.05 in this work.

### 3.2. Reinforcement Learning

During RL training, we optimize the policy on a mixture of two sample types: (i) standard on-policy rollouts $\tau \sim \pi_\theta(\cdot \mid q)$, and (ii) the revised trajectory generated under hindsight experience guidance

$\tilde{\tau}^\star \sim \pi_\theta(\cdot \mid q, \mathcal{H}_{i^\star})$. The policy learns from both the original sampling distribution and rubric-guided improvements. Our training objective is presented in the following:

$$\mathcal{J}_{\text{HeRL}}(\theta) = \mathbb{E}_{q \sim \mathcal{Q}, \{\tau_i\}_{i=1}^N \sim \pi_{\text{old}}(\cdot \mid q), \tilde{\tau}^\star \sim \pi_{\text{old}}(\cdot \mid q, \mathcal{H}_{i^\star})} \Bigg[ \frac{1}{N+1}$$

$$\left( \sum_{i=1}^N \frac{1}{|\tau_i|} \sum_{t=1}^{|\tau_i|} \text{CLIP}(\rho_{i,t}, \epsilon, A_i) + \frac{1}{|\tilde{\tau}^\star|} \sum_{t=1}^{|\tilde{\tau}^\star|} f(\tilde{\rho}_t) \, \tilde{A} \right) \Bigg], \quad (9)$$

where $\text{CLIP}(\rho, \epsilon, A) = (\rho \cdot A, \text{clip}(\rho, 1 - \epsilon, 1 + \epsilon) \cdot A)$, the advantages $A_i$ and $\tilde{A}$ are computed using the group mean with standard deviation of rewards from both the original and revised trajectories:

$$\mathcal{R}_G = \{r_i\}_{i=1}^N \cup \{\tilde{r}_{i^\star}\},$$

$$A_i = \frac{r_i - \text{mean}(\mathcal{R}_G)}{\text{std}(\mathcal{R}_G)}, \quad \tilde{A} = \frac{\tilde{r}_{i^\star} - \text{mean}(\mathcal{R}_G)}{\text{std}(\mathcal{R}_G)}. \quad (10)$$

The importance sampling ratio between the current policy and the old policy is define as:

$$\rho_{i,t} = \frac{\pi_\theta(\tau_{i,t}|q, \tau_{i,<t})}{\pi_{\text{old}}(\tau_{i,t}|q, \tau_{i,<t})}, \tilde{\rho}_t = \frac{\pi_\theta(\tilde{\tau}_t^\star|q, \mathcal{H}_{i^\star}, \tilde{\tau}_{<t}^\star)}{\pi_{\text{old}}(\tilde{\tau}_t^\star|q, \mathcal{H}_{i^\star}, \tilde{\tau}_{<t}^\star)}. \quad (11)$$

To enhance learning from low-probability tokens in revised trajectories, we further introduce policy shaping via regularized importance sampling (Yan et al., 2026):

$$f(x) = \frac{x}{x + \gamma}, \quad (12)$$

where we set $\gamma = 1$ in all experiments.

Note that the we preserve hindsight experience when calculating the importance sampling ratio, while masking the loss for the tokens of hindsight experience during training. The masking mechanism ensures the model learns to generate rubric-satisfying improvements *conditioned on* the hindsight experience, rather than learning to reproduce the feedback text or the original trajectory itself.

Overall, HeRL preserves the standard RLVR objective: the policy gradient is computed only over self-generated tokens, but operates over a richer trajectory structure that includes explicit experiential guidance. This allows the policy to reuse prior attempts to produce subsequent improvements, thereby reducing reliance on blind exploration.

### 3.3. Theoretical Analysis

In this section, we theoretically establish a connection between rubric satisfaction and the estimation of policy gradient to demonstrate the effectiveness of our method. The full proof can be found in Appendix D.

**Proposition 3.1.** *Let $w$ denote the point weight set of all rubrics, $w_+$ denote the point weight set of satisfied rubrics, and $w_-$ denote the point weight set of unsatisfied rubrics. The expected (ideal) reward is $R_I(\tau; q) = w^\top \cdot \mathbf{1}$, and the estimated reward at step $T$ is $R_T(\tau; q) = w_+^\top \cdot \mathbf{1}$. Assuming GRPO style updates, the gradient under expected reward and estimated reward can be expressed as $g_{R_I} = \mathbb{E}[\nabla_\theta \log \pi_\theta(\tau|q) R_I(\tau; q)]$ and $g_{R_T} = \mathbb{E}[\nabla_\theta \log \pi_\theta(\tau|q) R_T(\tau; q)]$, respectively. Then we have:*

$$\|g_{R_I} - g_{R_T}\|_2 \le \sqrt{\mathbb{E}\left[\left\|\nabla_\theta \log_{\pi_\theta}\right\|^2\right]} \|w_-\|_1. \quad (13)$$

*Remark* 3.2. Proposition 3.1 reveals that the difference between the ideal and estimated gradient is upper-bounded by $\|w_-\|_1$ times the expected squared norm of the policy score function. Augmenting the responses to better satisfy the given rubrics leads to better estimation of the ideal gradient, thereby enhancing the stability and efficiency during training. This provides a theoretical explanation for the effectiveness of HeRL: by tightening the upper bound on unmet rubrics, it yields a more reliable gradient direction and, in turn, improves learning efficiency and final performance.

## 4. Experiments

Our experiments seek to answer the following research questions: (1) Does HeRL deliver consistent performance gains over baselines, including SFT, DPO, and RLVR? (Table 1) (2) Whether HeRL improves task performance without sacrificing out-of-domin generalization? (Table 3) (3) Can HeRL extend the model's effective capability boundary and enable experience-guided self-improvement at test time? (Figure 4) (4) How effectively HeRL sustains exploration throughout training relative to the RLVR? (Figure 5)

### 4.1. Experimental Setup

**Datasets and Benchmark.** To assess whether HeRL improves LLMs' general reasoning capabilities, we evaluate it on three reasoning domains. **(1) instruction-following** (IFEval (Zhou et al., 2023), IFBench (Zhang et al., 2025e), and MulDimIF (Ye et al., 2025)), **(2) writing** (Writing-Bench (Wu et al., 2026)), and **(3) medicalQA** (LLMEval-Med (Zhang et al., 2025d) and HealthBench-500 (Arora et al., 2025)). We train with two data sources: instruction-following capabilities are learned from HIR-16K (Zhang et al., 2025c), while medical capability is trained on the full RaR-Medicine dataset (Gunjal et al., 2025), preserving its original rubrics. We do not train on writing tasks, as they resemble instruction-following and thus can probe HeRL's close-domain generalization capabilities. Moreover, we evaluate HeRL on out-of-distribution (OOD) datasets, including MATH-500 (Lightman et al., 2024), GPQA (Rein et al., 2024), and MMLU-Pro (Wang et al., 2024). The detailed dataset information is provided in Appendix C.

**Baselines and Evaluation Metrics.** We compare HeRL with three baselines. (1) SFT: for the instruction-following and writing subsets, we synthesize targets using GPT-5, while for RAR-Medicine we directly use the dataset-provided references. (2) DPO (Rafailov et al., 2023): we build preference pairs by using Qwen2.5-7B-Instruct (Team et al., 2024) to generate rejected responses and GPT-5 to produce the corresponding chosen responses. (3) RLVR (Lambert et al., 2024; Guo et al., 2025a): we optimize verifiable rewards defined at the rubric level, with the final scalar reward computed by explicit aggregation as in Eq (4). Detailed training hyperparameters are in Appendix A.

*Table 1.* Experimental results (%) across diverse domains using different LLMs trained on the same data. Best results among all methods are marked as **bolded**, and arrows indicate improvement or degradation over the initial model. Note Writing* tasks are not in training data.

| Method | Instruction Following | | | Writing* | Medical | |
|---|---|---|---|---|---|---|
| | **IFEval** | **IFBench** | **MulDimIF** | **WritingBench** | **LLMEval-Med** | **HealthBench-500** |
| Qwen2.5-7B-Instruct | 72.6 ↑0.0 | 26.2 ↑0.0 | 51.4 ↑0.0 | 57.0 ↑0.0 | 56.0 ↑0.0 | 24.4 ↑0.0 |
| + SFT | 75.6 ↑3.0 | 27.9 ↑1.7 | 67.8 ↑16.4 | 51.5 ↓5.5 | 34.8 ↓21.2 | 27.2 ↑2.8 |
| + DPO | 66.9 ↓5.7 | 25.9 ↓0.3 | 56.5 ↑5.1 | 52.1 ↓4.9 | 35.8 ↓20.2 | 28.0 ↑3.6 |
| + RLVR | 77.3 ↑4.7 | 31.6 ↑5.4 | 73.5 ↑22.1 | 54.8 ↓2.2 | 60.5 ↑4.5 | 30.5 ↑6.1 |
| **+ HeRL (Ours)** | **82.4** ↑9.8 | **39.7** ↑13.5 | **83.4** ↑32.0 | **59.1** ↑2.1 | **65.0** ↑9.0 | **34.3** ↑9.9 |
| Llama-3.2-3B-Instruct | 71.2 ↑0.0 | 23.8 ↑0.0 | 35.8 ↑0.0 | 30.5 ↑0.0 | 16.1 ↑0.0 | 14.5 ↑0.0 |
| + SFT | 73.0 ↑1.8 | 24.8 ↑1.0 | 66.9 ↑31.1 | 24.5 ↓6.0 | 15.1 ↓1.0 | 21.0 ↑6.5 |
| + DPO | 74.3 ↑3.1 | 22.1 ↓1.7 | 54.4 ↑18.6 | 14.4 ↓16.1 | 11.5 ↓4.6 | 13.5 ↓1.0 |
| + RLVR | 79.1 ↑7.9 | 26.6 ↑2.8 | 77.6 ↑41.8 | 39.7 ↑9.2 | 18.5 ↑2.4 | 17.8 ↑3.3 |
| **+ HeRL (Ours)** | **82.4** ↑11.2 | **30.6** ↑6.8 | **84.7** ↑48.9 | **45.4** ↑14.9 | **18.7** ↑2.6 | **26.6** ↑12.1 |
| Qwen3-4B-Instruct-2507 | 83.4 ↑0.0 | 29.9 ↑0.0 | 57.3 ↑0.0 | 84.3 ↑0.0 | 74.5 ↑0.0 | 42.0 ↑0.0 |
| + SFT | 83.4 ↑0.0 | 31.3 ↑1.4 | 66.8 ↑9.5 | 81.7 ↓2.6 | 73.3 ↓1.2 | 36.0 ↓6.0 |
| + DPO | 83.9 ↑0.5 | 27.9 ↓2.0 | 61.5 ↑4.2 | 85.0 ↑0.7 | 74.9 ↑0.4 | 39.1 ↓2.9 |
| + RLVR | 85.8 ↑2.4 | 36.9 ↑7.0 | 79.0 ↑21.7 | 83.9 ↓0.4 | 78.1 ↑3.6 | 41.7 ↓0.3 |
| **+ HeRL (Ours)** | **86.1** ↑2.7 | **39.7** ↑9.8 | **82.5** ↑25.2 | **85.7** ↑1.4 | **79.3** ↑4.8 | **43.6** ↑1.6 |

**Models and Configurations.** We train and compare models on different architectures and scales, including Qwen2.5-7B-Instruct (Team et al., 2024), Llama-3.2-3B-Instruct (Meta, 2024) , Qwen3-4B-Instruct-2507 (Yang et al., 2025). We implement RLVR and our HeRL algorithm on top of the verl (Sheng et al., 2025) training framework, and use LLaMA-Factory (Zheng et al., 2024) for SFT and DPO training. For RLVR, we sample 8 responses per instruction; for HeRL, we sample 7 rollouts and generate 1 hindsight experience guided trajectory. All rubric evaluations are performed by GPT-4o mini (Hurst et al., 2024), which serves as the judge model in our experiments. The full evaluation prompt used for this process is provided in Appendix F.

### 4.2. Main Results

**HeRL outperforms existing baselines.** As shown in Table 1, HeRL demonstrates strong performance across model families and scales. Specifically, based on Qwen2.5-7B-Instruct, Llama-3.2-3B-Instruct and Qwen3-4B-Instruct-2507, HeRL achieves the best results across all six benchmarks. Compared with standard RLVR, HeRL delivers further improvements, indicating that hindsight-guided revisions provide a stronger driver for exploration. In contrast to SFT and DPO, HeRL is also less prone to performance degradation, yielding more stable performance overall. Notably, although our training data does not include any writing tasks, HeRL still improves on WritingBench, whereas other baselines more often suffer cross-domain drops.

**HeRL maintains OOD reasoning ability.** To further assess whether HeRL could hurt out-of-distribution (OOD) generalization, we additionally evaluate the model on three

*Table 2.* Performance of HeRL on out-of-domain benchmarks.

| Model | MATH-500 | GPQA | MMLU-Pro |
|---|---|---|---|
| Llama-3.2-3B-Instruct | 38.6 ↑0.0 | 26.5 ↑0.0 | 33.9 ↑0.0 |
| + RLVR | 39.6 ↑1.0 | 29.2 ↑2.7 | 35.1 ↑1.2 |
| **+ HeRL (Ours)** | 36.6 ↓2.0 | 29.2 ↑2.7 | 34.8 ↑0.9 |
| Qwen2.5-7B-Instruct | 76.0 ↑0.0 | 34.1 ↑0.0 | 57.3 ↑0.0 |
| + RLVR | 74.6 ↓1.4 | 34.3 ↑0.2 | 54.3 ↓3.0 |
| **+ HeRL (Ours)** | 77.6 ↑1.6 | 33.9 ↓0.2 | 55.5 ↓1.8 |
| Qwen3-4B-Instruct-2507 | 86.0 ↑0.0 | 46.6 ↑0.0 | 61.5 ↑0.0 |
| + RLVR | 88.0 ↑2.0 | 43.0 ↓3.6 | 61.4 ↓0.1 |
| **+ HeRL (Ours)** | 89.8 ↑3.8 | 45.3 ↓1.3 | 62.1 ↑0.6 |

OOD reasoning benchmarks that are only weakly related to medical or instruction-following tasks. As shown in the Table 2, despite being trained solely on medical and instruction-following data, the HeRL-trained model performs comparably to the original base model on these OOD benchmarks, showing no obvious collapse or generalization drop. These results indicate that HeRL improves exploration efficiency under rubric-based training while preserving broad reasoning capability and cross-domain robustness.

**HeRL improves sampling efficiency and reasoning ability boundary.** Beyond Pass@1, we evaluate HeRL and RLVR under increasing sampling budgets $k$ (Pass@$k$). As shown in Figure 4 (a), HeRL outperforms RLVR across all $k$, achieves stronger performance at small $k$ (higher sampling efficiency), and maintains its advantage as $k$ grows, indicating a higher attainable performance.

**HeRL can be further scaled at test time under experience guided self-improvement.** To evaluate whether hindsight experience guidance for exploration in training

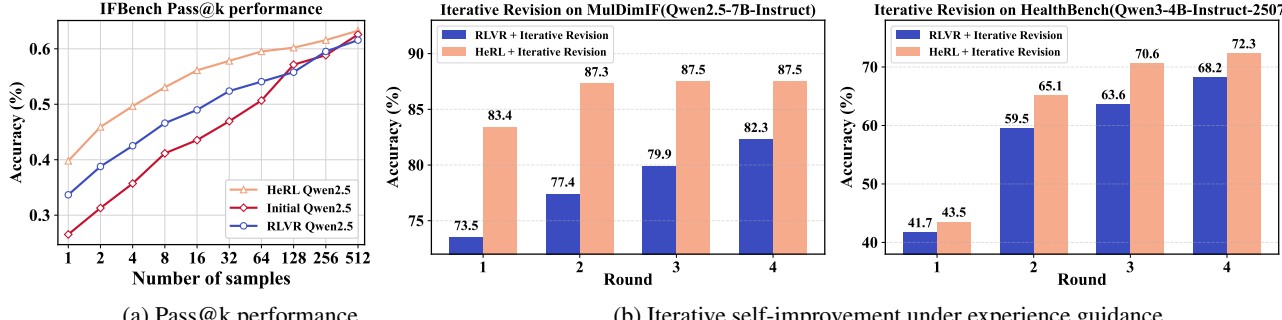

(a) Pass@k performance.                    (b) Iterative self-improvement under experience guidance.

*Figure 4.* Investigation of HeRL's sampling efficiency. (a) Pass@k performance of Qwen2.5-7B-Instruct, RLVR, and HeRL on IFBench. (b) HeRL outperforms RLVR under iterative revision with experience guidance on HealthBench-500 with Qwen3-4B-Instruct-2507 and MulDimIF with Qwen2.5-7B-Instruct.

*Table 3.* Ablation study of HeRL, where NaiveHE denotes naive Hindsight Experience, HE refers to Hindsight Experience, and BR is Bonus Reward. The arrows indicate the performance gains.

| Setting | IFBench | WritingBench | HealthBench-500 |
|---|---|---|---|
| *Model I: Qwen2.5-7B-Instruct* | | | |
| Baseline (RLVR) | 31.6 ↑0.0 | 54.8 ↑0.0 | 30.5 ↑0.0 |
| + NaiveHE | 32.3 ↑0.7 | 52.7 ↓2.1 | 28.8 ↓1.7 |
| + HE | 36.7 ↑5.1 | 58.9 ↑4.1 | 31.8 ↑1.3 |
| + HE + BR(**HeRL**) | **39.7** ↑8.1 | **59.1** ↑4.3 | **34.3** ↑3.8 |
| *Model II: Llama-3.2-3B-Instruct* | | | |
| Baseline (RLVR) | 26.6 ↑0.0 | 39.7 ↑0.0 | 17.8 ↑0.0 |
| + NaiveHE | 28.3 ↑1.7 | 33.7 ↓6.0 | 23.9 ↑6.1 |
| + HE | 28.1 ↑1.5 | 41.1 ↑1.4 | 23.5 ↑5.7 |
| + HE + BR(**HeRL**) | **30.6** ↑4.0 | **45.4** ↑5.7 | **26.6** ↑8.8 |
| *Model III: Qwen3-4B-2507-Instruct* | | | |
| Baseline (RLVR) | 36.9 ↑0.0 | 83.9 ↑0.0 | 41.7 ↑0.0 |
| + NaiveHE | 35.7 ↓1.2 | 66.2 ↓17.7 | 40.9 ↓0.8 |
| + HE | 37.7 ↑0.8 | 85.3 ↑1.4 | 43.8 ↑2.1 |
| + HE + BR(**HeRL**) | **39.7** ↑2.8 | **85.7** ↑1.8 | **43.6** ↑1.9 |

time transfers to test time, we examine whether the model can iteratively improve its responses under guidance during inference. We conduct iterative guided revision on Health-Bench. Starting from Round 2, at each sampling round, we construct hindsight experience using the original trajectory and unmet rubrics for guidance as in training. The hindsight experience is fed to the model to generate a revised trajectory for the next round. As shown in Figure 4 (b), HeRL with iterative guided revision outperforms RLVR , suggesting that hindsight experience guidance learned in training transfers to more effective revision during inference. Moreover, HeRL achieves a larger improvement when guidance is introduced, suggesting that this inference-time scaling capability does not stem solely from the model's in-context learning ability, but also benefits from the experience-guided revision capability internalized during training.

## 4.3. Ablation Study

To validate the effectiveness of our two key components—*Hindsight Experience* (**HE**) and *Bonus Reward*

(**BR**), we conduct ablation studies on three representative datasets from different domains. We compare four settings: **(1) RLVR**, the baseline; **(2) + NaiveHE**, which directly adds the original trajectory with the revised response and trains on $(q, \tilde{\tau}^\star)$ without preserving the hindsight experience; **(3) + HE**, which preserves hindsight experience while masking its loss in training; and **(4) + HE + BR**, i.e., full HeRL with bonus rewards to incentivize responses with higher improvement potential under guidance. As shown in Table 3, unlike NaiveHE, which directly reuses revised responses, standard HE consistently improves performance, indicating that retaining hindsight experience is critical for optimization process. Notably, NaiveHE even leads to performance drops in some cases, likely because it treats revised samples as off-policy training data, which introduces training instability. Adding BR on top of HE yields further performance gains, suggesting that the bonus reward improves exploration and thereby leads to better overall performance.

*Table 4.* Ablation study on the number of hindsight experience samples . HE=1 and HE=2 denote using one or two hindsight-experience guided revised trajectories per group. The arrows indicate the performance gains relative to the RLVR baseline. Full results across all six benchmarks are provided in Appendix B.

| Setting | IFBench | WritingBench | HealthBench-500 |
|---|---|---|---|
| *Model: Qwen2.5-7B-Instruct* | | | |
| RLVR | 31.6 ↑0.0 | 54.8 ↑0.0 | 30.5 ↑0.0 |
| HE=2 | 38.5 ↑6.9 | 56.0 ↑1.2 | 33.2 ↑2.7 |
| HE=1(**HeRL**) | **39.7** ↑8.1 | **59.1** ↑4.3 | **34.3** ↑3.8 |

**Hindsight Experience Numbers.** Table 4 compares different numbers of hindsight experience samples. Both HE=1 and HE=2 improve over RLVR, showing that guided revision trajectories are useful. However, HE=2 does not bring consistent gains over HE=1, suggesting that more HE samples may introduce additional off-policy bias. Therefore, we use HE=1 during training.

Additional experiments are provided in Appendix B, where we report the full results across all six benchmarks and analyze other design choices of HeRL.

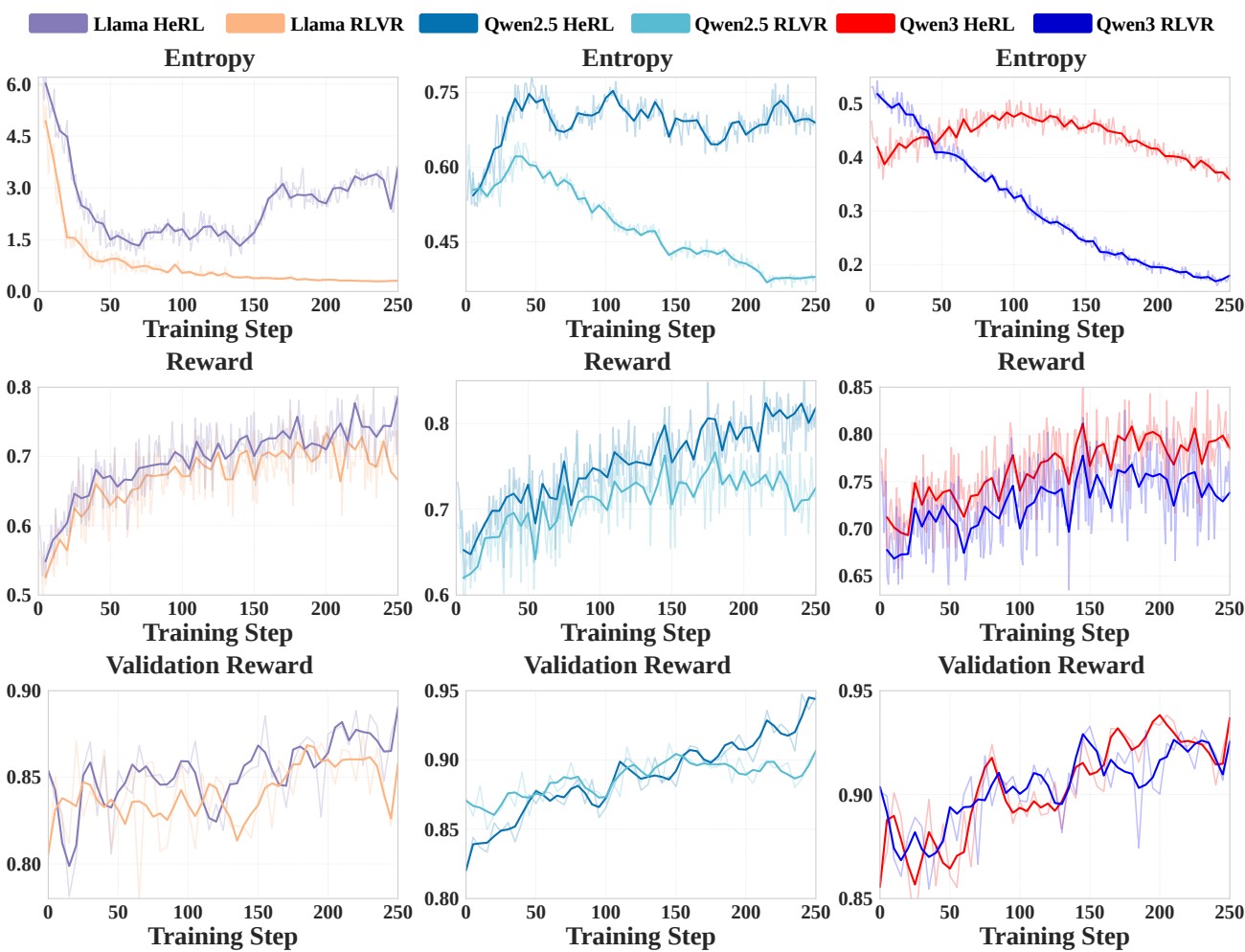

*Figure 5.* Training dynamics of different models. (Top) Entropy, (Mid) Reward and (Bottom) Validation Reward curves over training steps on the RAR-Medicine dataset are reported for three base models, comparing HeRL with RLVR baseline. Model names are abbreviated in the plots: Qwen2.5 denotes Qwen2.5-7B-Instruct, Qwen3 denotes Qwen3-4B-Instruct-2504, and Llama denotes Llama3.2-3B-Instruct.

## 4.4. Analysis

**Training Dynamics.** As shown in Figure 5, HeRL exhibits more effective and stable training dynamics across different models. In the top row, HeRL maintains consistently higher sampling entropy throughout training. In contrast, RLVR shows an early drop in entropy and stays at a low-entropy regime, indicating an early contraction of the sampling distribution. By keeping higher entropy, HeRL avoids premature contraction and continues to generate diverse candidates with improvement potential. Consistent with this pattern, the mid and bottom rows show that HeRL achieves higher rewards than RLVR, attains higher validation rewards, and retains a clearer advantage in later training.

## 5. Related Work

**Exploration in LLM Reinforcement Learning.** Although reinforcement learning has significantly improved LLM's problem-solving capabilities, they often suffer from diversity collapse(Song et al., 2025). Effective exploration is therefore crucial to prevent the policy from being trapped in local optima. Recent works have taken two main directions. The first focuses on **structured exploration**; methods such as TreeRL(Hou et al., 2025) and RFTT(Zhang et al., 2025a) leverage tree-based search structures to systematically explore the reasoning space. FR3E(Zheng et al., 2025) first identifies high-uncertainty decision points and then performs targeted rollouts instead of full rollouts. RuscaRL (Zhou et al., 2025) injects the rubrics into instruction, but it still suffers from ineffective guidance and off-policy updates. The second line of research leverages **intrinsic rewards**. Various signals such as solution diversity(Yao et al., 2026), curiosity(Dai et al., 2025; Zhang et al., 2025b), entropy(Cheng et al., 2026), and count-based reward(Zhang et al., 2025f) are used to encourage unconventional but potentially optimal trajectories. However, current methods primarily focus on improving the efficiency and diversity

of exploration, yet rarely uncover trajectories that exceed the capabilities of the underlying model. In contrast, HeRL leverages Hindsight Experience to guide the policy toward exploring desired responses that lie beyond the current data distribution.

**LLM Reinforcement Learning with Rubric Reward.** Reinforcement learning with verifiable reward signals (RLVR) has become a prominent paradigm for enhancing large language models (LLMs). Recent efforts have demonstrated effective verification mechanisms in a range of domains, including mathematical reasoning (Guo et al., 2025a; Lambert et al., 2024; Wang et al., 2026), code (Rastogi et al., 2025; Liu et al., 2025; Wen et al., 2025) and instruction following (Guo et al., 2025b; Qin et al., 2026; Peng et al., 2025). However, applying RLVR to open-ended reasoning tasks remains difficult, as they cannot be evaluated by discrete, rule-defined criteria. Recent work has extended RLVR to rubric-based rewards to checklist-style judgements generated by LLM. Pioneering approaches such as RaR (Gunjal et al., 2025) and Huang et al. (2025b) have applied this framework to general reasoning domains, while AdvancedIF (He et al., 2025) employs it for instruction following. In contrast to these methods, which use rubric-based rewards primarily as reward signals for RL, we further leverage them to guide exploration to extend the boundary of model's capability.

## 6. Conclusion

This work proposes HeRL, a framework designed to bootstrap *effective exploration* in RL for LLM reasoning guided by hindsight experience. Leveraging the language description from rubrics and in-context learning abilities of LLMs, HeRL enables the model to efficiently generate desired high-quality responses for learning without blind trial-and-error from scratch. To further stabilize the process of exploration and exploitation, we introduce a bonus reward to incentivize responses with greater improvement potential, and a policy shaping mechanism to effectively learn from unfamiliar but desirable improvements. Extensive experiments demonstrate that HeRL consistently outperforms strong baselines and can further benefit from experience-guided self-improvement at test time.

While HeRL demonstrates clear gains in exploration efficiency and final performance, it has several limitations. First, it depends on high-quality, broad-coverage rubric datasets, which remain scarce and expensive to create. Second, the rubrics are predefined and static. Since the model's capability boundary evolves over time, fixed rubrics may not always target the most informative learning gap. Future work may explore adaptive rubrics that evolve with training, for example by adjusting difficulty based on the model's weaknesses, to further improve sample efficiency and stability.

## Acknowledgements

This work is supported by the Strategic Priority Research Program of the Chinese Academy of Sciences under Grant No. XDB1020201.

## Impact Statement

This paper presents work whose goal is to advance the exploration in reinforcement learning for LLM reasoning. There are many potential societal consequences of our work, none of which we feel must be specifically highlighted here.

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

# A. Implementation Details

All experiments run on 8×A100-80GB GPUs. We use LLaMA-Factory (Zheng et al., 2024) for SFT and DPO training, and verl (Sheng et al., 2025) framework for RL training. Detailed training configurations of SFT and DPO are presented in Table 5a, and the training configurations of RL are presented in Table 5b.

*Table 5.* Training configurations across different methods and model backbones.

*(a)* Training configuration of SFT and DPO.

| Method | SFT, DPO |
|---|---|
| **Training** | per_device_train_batch_size = 16, gradient_accumulation_steps = 16 |
| | learning_rate = 1e-6, lr_scheduler_type = constant |
| | cutoff_len = 4096, warmup_steps = 10, epochs = 5 |
| **Optimizations** | deepspeed: z3, bf16 |

*(b)* Training configuration of RL.

| Settings | Hyperparameters |
|---|---|
| **HeRL** | rollout_n = 7, Hindsight experience = 1, $\alpha$ = 0.05 |
| **RLVR** | rollout_n = 8 |
| **Sampling** | top_k = -1, top_p = 1.0, temperature = 1.0 |
| | max_prompt_length = 2,048, max_response_length = 4,096 |
| **Training** | ppo_mini_batch_size = 64, ppo_micro_batch_size_per_gpu = 8 |
| | log_prob_micro_batch_size_per_gpu = 8 |
| | learning_rate = 1e-6, kl_loss_coef = 1e-4, epochs = 5 |
| **Optimizations** | param_offload, flash_attn, bf16 |

# B. Additional Results

This section reports additional studies on Qwen2.5-7B-Instruct to further analyze several design choices in HeRL. These studies examine the number of hindsight experience samples, whether revision trajectories can be used as SFT data, whether source trajectory tokens should be optimized, whether the bonus reward alone is sufficient, and whether the improvement comes from judge distillation.

*Table 6.* Ablation study on the number of hindsight experience samples on Qwen2.5-7B-Instruct. HE=1 and HE=2 denote using one or two hindsight experience guided revised trajectories per group, respectively. Arrows indicate changes relative to RLVR.

| Setting | IFEval | IFBench | MulDimIF | WritingBench | LLMEval-Med | HealthBench-500 |
|---|---|---|---|---|---|---|
| RLVR | 77.3 ↑0.0 | 31.6 ↑0.0 | 73.5 ↑0.0 | 54.8 ↑0.0 | 60.5 ↑0.0 | 30.5 ↑0.0 |
| HE=2 | 81.9 ↑4.6 | 38.5 ↑6.9 | 82.7 ↑9.2 | 56.0 ↑1.2 | 61.9 ↑1.4 | 33.2 ↑2.7 |
| HE=1 (**HeRL**) | **82.4** ↑5.1 | **39.7** ↑8.1 | **83.4** ↑9.9 | **59.1** ↑4.3 | **65.0** ↑4.5 | **34.3** ↑3.8 |

## B.1. Can Revision Trajectories Be Used as SFT Data?

We test this alternative by collecting revision trajectories on the training set and using them for supervised fine-tuning ("SFT w/ Revision"). The results in Table 7 show that this does not recover the benefit of HeRL: SFT Revision is not only much weaker than HeRL, but also underperforms standard SFT on IFEval, IFBench, and MulDimIF. This suggests that the gain of HeRL is not from simply adding revised responses as extra SFT data. We believe the reason is that revision trajectories are biased and heterogeneous supervision generated from failed attempts; direct SFT treats them as uniformly correct targets, which can hurt general instruction-following. In contrast, HeRL uses them within reward-guided RL training rather than pure imitation. We do observe relatively larger gains on medically oriented evaluation, especially LLMEval-Med, likely because medical rubrics are more fine-grained and thus provide richer supervision.

*Table 7.* Comparison of using revision trajectories as supervised fine-tuning data on Qwen2.5-7B-Instruct. Arrows indicate changes relative to the base model.

| Setting | IFEval | IFBench | MulDimIF | WritingBench | LLMEval-Med | HealthBench-500 |
|---|---|---|---|---|---|---|
| Qwen2.5-7B-Instruct | 72.6 ↑0.0 | 26.2 ↑0.0 | 51.4 ↑0.0 | 57.0 ↑0.0 | 56.0 ↑0.0 | 24.4 ↑0.0 |
| + SFT | 75.6 ↑3.0 | 27.9 ↑1.7 | 67.8 ↑16.4 | 51.5 ↓5.5 | 34.8 ↓21.2 | 27.2 ↑2.8 |
| + SFT w/ Revision | 70.3 ↓2.3 | 27.1 ↑0.9 | 52.4 ↑1.0 | 53.2 ↓3.8 | 55.9 ↓0.1 | 27.6 ↑3.2 |
| **+ HeRL** | **82.4** ↑9.8 | **39.7** ↑13.5 | **83.4** ↑32.0 | **59.1** ↑2.1 | **65.0** ↑9.0 | **34.3** ↑9.9 |

## B.2. Should We Train on Source Trajectory Tokens?

We examine whether the source trajectory tokens should be included in the optimization target. As shown in Table 8, applying loss to source trajectory tokens substantially weakens performance compared with HeRL and even falls below RLVR on most benchmarks. This suggests that the source trajectory introduces substantial noise rather than useful supervision in this hindsight context. Our interpretation is that the original trajetory mainly serves as guidance for identifying what failed and guiding the revision, but forcing the model to imitate those tokens also reinforces suboptimal generations.

*Table 8.* Ablation study on training with source trajectory loss on Qwen2.5-7B-Instruct. Arrows indicate changes relative to RLVR.

| Setting | IFEval | IFBench | MulDimIF | WritingBench | LLMEval-Med | HealthBench-500 |
|---|---|---|---|---|---|---|
| RLVR | 77.3 ↑0.0 | 31.6 ↑0.0 | 73.5 ↑0.0 | 54.8 ↑0.0 | 60.5 ↑0.0 | 30.5 ↑0.0 |
| HeRL w/ Source Trajectory Loss | 73.9 ↓3.4 | 29.9 ↓1.7 | 70.2 ↓3.3 | 57.8 ↑3.0 | 52.4 ↓8.1 | 30.4 ↓0.1 |
| **HeRL** | **82.4** ↑5.1 | **39.7** ↑8.1 | **83.4** ↑9.9 | **59.1** ↑4.3 | **65.0** ↑4.5 | **34.3** ↑3.8 |

## B.3. Is the Bonus Reward Alone Sufficient?

We isolate the effect of the bonus reward by applying it without hindsight experience guided revision trajectories. As shown in Table 9, BR-only improves over RLVR on most benchmarks, indicating that the bonus reward can help prioritize promising samples. However, it remains clearly weaker than full HeRL, especially on IFBench, MulDimIF, LLMEval-Med, and HealthBench-500. This shows that the bonus reward alone is helpful but insufficient; the strongest gains require both hindsight experience guided revision and reward-guided optimization.

*Table 9.* Ablation study on the bonus reward on Qwen2.5-7B-Instruct. BR-only denotes applying the bonus reward without hindsight experience guided revision trajectories. Arrows indicate changes relative to RLVR.

| Setting | IFEval | IFBench | MulDimIF | WritingBench | LLMEval-Med | HealthBench-500 |
|---|---|---|---|---|---|---|
| RLVR | 77.3 ↑0.0 | 31.6 ↑0.0 | 73.5 ↑0.0 | 54.8 ↑0.0 | 60.5 ↑0.0 | 30.5 ↑0.0 |
| RLVR w/ BR-only | 80.1 ↑2.8 | 33.6 ↑2.0 | 75.7 ↑2.2 | 57.9 ↑3.1 | 62.6 ↑2.1 | 28.3 ↓2.2 |
| **HeRL** | **82.4** ↑5.1 | **39.7** ↑8.1 | **83.4** ↑9.9 | **59.1** ↑4.3 | **65.0** ↑4.5 | **34.3** ↑3.8 |

## B.4. Does HeRL Benefit from Judge Distillation?

*Table 10.* Comparison under different judge models used during training on Qwen2.5-7B-Instruct. Arrows indicate changes relative to the base model.

| Method | Judge | HealthBench-500 | LLMEval-Med |
|---|---|---|---|
| Qwen2.5-7B-Instruct | – | 24.4 ↑0.0 | 56.0 ↑0.0 |
| +RLVR | Qwen3-8B | 27.5 ↑3.1 | 53.5 ↓2.5 |
| +HeRL | Qwen3-8B | 30.3 ↑5.9 | 58.4 ↑2.4 |
| +RLVR | GPT-4o-mini | 30.5 ↑6.1 | 60.5 ↑4.5 |
| +HeRL | GPT-4o-mini | **34.3** ↑9.9 | **65.0** ↑9.0 |

We examine whether HeRL's improvement comes from distilling the judge or the unmet rubric. Several observations suggest that this is not the case. First, the loss on rubric tokens is masked, so the model is not trained to imitate the rubric text. The unmet rubrics are only used as conditioning information for revision. Second, Table 7 shows that directly using revision

trajectories for SFT remains much weaker than HeRL, indicating that the benefit cannot be explained by supervised imitation of revised answers. Third, we replace GPT-4o-mini with a weaker judge, Qwen3-8B. As shown in Table 10, both RLVR and HeRL degrade under the weaker judge. If the main mechanism were judge distillation, a weaker judge would not be expected to provide a stronger signal. Instead, the degradation suggests that lower-quality criterion-level judgments introduce noisier rewards and less reliable hindsight experience. This supports our interpretation that HeRL benefits from accurate rubric-level evaluation and guided exploration, rather than from judge distillation.

## C. Dataset

### C.1. Training Dataset

We select two rubrics datasets to train HeRL and to validate the effectiveness of hindsight experience: HIR-16K and RAR-Medicine. Both datasets provide fine-grained, verifiable evaluation criteria.

**HIR-16K (Zhang et al., 2025c)** is designed specifically for hindsight revising with decomposable, multi-constraint instructions. It contains 16,969 queries collected from public instruction-following sources, where each instruction is decomposed into atomic constraints and filtered to retain sufficiently complex cases. The final dataset includes 76,456 hard constraints and 46,536 soft constraints, supporting training and evaluation under dense constraint settings.

**RAR-Medicine (Gunjal et al., 2025)** is a medical reasoning dataset comprising 20K prompts curated from diverse medical reasoning sources, including medical-o1-reasoning-SFT, Natural Reasoning, SCP-116K, and GeneralThought-430K. To enable reliable scoring, RAR-Medicine provides instance-specific rubrics, which are automatically generated using GPT-4o, allowing each response to be evaluated against prompt-specific criteria.

### C.2. Evaluation Dataset

**IFEval (Zhou et al., 2023)** is a benchmark designed to measure how well models can strictly follow detailed instructions. It contains 541 instruction instances spanning 25 categories of verifiable constraints. Each prompt includes one or more verifiable instructions, and the evaluation relies on rubric-style checks such as required keyword inclusion and keyword frequency, alongside other constraint satisfaction criteria.

**IFBench (Zhang et al., 2025e)** introduces 58 new, diverse, and challenging verifiable constraints, and provides 294 instruction instances in total. Its rubrics emphasize precise compliance, including checks like output formatting requirements and word-count or length-related constraints, enabling automatic and objective evaluation of instruction-following robustness beyond seen constraint types.

**MuIDimIF (Ye et al., 2025)** performs constraint expansion, conflict detection, and instruction rewriting, producing 1,200 code-verifiable samples. The resulting rubrics cover a broad range of constraints, including format, language selection, and length control, with programmatic validators used to determine whether the model output satisfies requirements.

**WritingBench (Wu et al., 2026)** is a comprehensive benchmark for assessing LLM writing performance across 6 core writing domains and 100 subdomains. It contains 1,000 instructions paired with detailed scoring rubrics. In addition to fixed rubrics, WritingBench proposes a query-dependent evaluation framework that allows an LLM to dynamically generate instance-specific assessment criteria tailored to each prompt.

**LLMEval-Med (Zhang et al., 2025d)** is an open-source benchmark designed to measure both the performance and safety of LLMs in healthcare settings. It consists of 5,000 multi-turn conversations between a model and either an individual user or a healthcare professional, reflecting realistic, open-ended clinical interactions. Evaluation is based on 48,562 unique rubric criteria covering a wide range of healthcare contexts (e.g., acute and non-acute scenarios) and behavioral dimensions (e.g., accuracy, instruction adherence, and communication quality), enabling fine-grained assessment of both correctness and safety-relevant behavior.

**HealthBench (Arora et al., 2025)** is a medical benchmark that evaluates LLMs across multiple core medical competencies, including knowledge, understanding, reasoning, safety, and medical text generation. It contains 2,996 questions derived from real-world electronic health records and expert-designed clinical scenarios, aiming to reflect authentic clinical decision-making and documentation needs. It also provides an automated evaluation pipeline that incorporates expert-developed checklists into an LLM-as-Judge framework, improving scoring consistency by grounding judgments in explicit, domain-informed criteria.

# D. Proof

## D.1. Derivation of Eq. (2)

For the token $y_t$, its gradient simplifies to:

$$
\begin{aligned}
&\frac{\partial \log \pi_\theta(y_t \mid q, y_{<t}) A_t}{\partial z_v} \\
&= \frac{\partial \pi_\theta(y_t \mid q, y_{<t})}{\partial z_v} \cdot \frac{A_t}{\pi_\theta(y_t \mid q, y_{<t})} \\
&= \frac{\mathbb{I}(v = y_t) \exp(z_{y_t}) \sum_{v' \in \mathcal{V}} \exp(z_{v'}) - \exp(z_{y_t}) \exp(z_v)}{\left(\sum_{v' \in \mathcal{V}} \exp(z_{v'})\right)^2} \cdot \frac{A_t}{\pi_\theta(y_t \mid q, y_{<t})} \\
&= \begin{cases} \left(1 - \pi_\theta(y_t \mid q, y_{<t})\right) \cdot A_t & \text{if } v = y_t \quad \text{(sampled token)} \\ -\pi_\theta(v \mid q, y_{<t}) \cdot A_t & \text{otherwise} \quad \text{(unsampled token)} \end{cases}.
\end{aligned}
\tag{14}
$$

## D.2. Proof of Proposition 3.1

*Proof.* The discrepancy between the ideal gradient and the estimated gradient is:

$$
\begin{aligned}
g_{R_I} - g_{R_T} &= \mathbb{E}_{q \sim \mathcal{D}, \tau \sim \pi_\theta(\cdot|q)} \left[ \nabla_\theta \log \pi_\theta(\tau|q) \left( R_I - R_T \right) \right] \\
&= \mathbb{E}_{q \sim \mathcal{D}, \tau \sim \pi_\theta(\cdot|q)} \left[ \nabla_\theta \log \pi_\theta(\tau|q) \left( X - \mathbb{E}_{q \sim \mathcal{D}, \tau \sim \pi_\theta(\cdot|q)} [X] \right) \right],
\end{aligned}
$$

where $X = R_I - R_T$. This derivation is obtained by the fact that $\mathbb{E}_{q \sim \mathcal{D}, \tau \sim \pi_\theta(\cdot|q)} \left[ \nabla_\theta \log \pi_\theta(\tau|q) \right] = 0$, so we can center $X$ without changing the expectation. Then we have

$$
\begin{aligned}
\left\| g_{R_I} - g_{R_T} \right\|_2 &= \left\| \mathbb{E}_{q \sim \mathcal{D}, \tau \sim \pi_\theta(\cdot|q)} \left[ \nabla_\theta \log \pi_\theta(\tau|q) \left( X - \mathbb{E}_{q \sim \mathcal{D}, \tau \sim \pi_\theta(\cdot|q)} [X] \right) \right] \right\|_2 \\
&\leq \sqrt{\mathbb{E} \left[ \left\| \nabla_\theta \log_{\pi_\theta} \right\|^2 \right]} \sqrt{Var(X)} \qquad \text{by Cauchy-Schwarz} \\
&= \sqrt{\mathbb{E} \left[ \left\| \nabla_\theta \log_{\pi_\theta} \right\|^2 \right]} \sqrt{Var(R_I - R_T)} \\
&= \sqrt{\mathbb{E} \left[ \left\| \nabla_\theta \log_{\pi_\theta} \right\|^2 \right]} \sqrt{\mathbb{E} \left[ (R_I - R_T)^2 \right]} \\
&= \sqrt{\mathbb{E} \left[ \left\| \nabla_\theta \log_{\pi_\theta} \right\|^2 \right]} \left\| w_- \right\|_1.
\end{aligned}
$$

$\square$

# E. Prompt for Hindsight Experience Guided Exploration

Here is the revise prompt we use.

> **Rar-Medicine Revise Prompt Template**
>
> You are a revise assistant. Keep satisfied rubrics intact and fix the unmet ones. Avoid adding new facts.
> **Task.** Revise the draft to satisfy the instruction *and* the rubric status below.
> **Rules.**
> (1) Keep the content that already satisfies **Satisfied** rubrics unchanged as much as possible.
> (2) Make the **minimum necessary edits** to satisfy all **Unmet** rubrics.
> (3) Do not introduce new requirements beyond the rubric; do not add unsupported facts.
> (4) Preserve formatting requirements (e.g., structure, ordering, required keywords) if they are listed as satisfied.
> (5) Output **only** the revised answer.

**[Instruction]**
{instruction}

**[Rubric Status]**
**Satisfied:**
{satisfied_block}
**Unmet:**
{unmet_block}

**[Draft]**
{draft}

**Return only the revised answer.**

---

**IF-Dataset Revise Prompt Template**

You are a revise assistant. Keep all already satisfied rubrics intact and fix the unmet ones. Avoid adding new facts.
**Task.** Revise the draft to satisfy the instruction *and* the Candidate status below.
**Rules.**
(1) Keep the content that already satisfies **Satisfied** rubrics unchanged as much as possible.
(2) Make the **minimum necessary edits** to satisfy all **Unmet** rubrics.
(3) Preserve formatting requirements (e.g., structure, ordering, required keywords) if they are listed as satisfied.
(4) Output **only** the revised answer.

**[Instruction]**
{instruction}

**[Candidate]**
**Satisfied:**
{satisfied_block}
**Unmet:**
{unmet_block}

**[Draft]**
{draft}

**Return only the revised answer.**

## F. Prompt for Evaluation

**Evaluation Prompt Template**

**[SYSTEM]**
You are an expert evaluator. For each rubric item, decide whether the response satisfies it. Return a JSON object with a single key "results" whose value is a list of 0/1 integers. The list length must equal the number of rubrics and each entry corresponds to the rubric with the same index.

**[USER]**
Evaluate the response against each rubric independently.

**[prompt]**
{prompt}

**[response]**
{response}

**[rubrics]**
{rubrics}

Return only JSON: {"results":  [0 or 1, ...]}

# G. Case Study

Here we provide detailed revise case during training.

---

**Instruction from MulDimIF**

Are piano chords utilized in house and EDM music, and how do they compare?
The output must follow the following rules:
1. The answer must include headings for each genre: 'House' and 'EDM'.
2. The answer must include a table with a maximum of three columns to compare the use of piano chords in house and EDM.
3. The answer must be between 150 and 200 words.
4. The answer must contain between 8 and 12 sentences.
5. The answer must include the keyword 'synthesizer'.
6. The answer must end with a question mark.
7. The answer must use capitalized letters for each word.

---

**Original response from Qwen2.5-7B-Instruct**

| Genre | Instrument | Chord Type |
|-------|------------|------------|
| House | Piano | Major |
| Edm | Synthesizer | Minor |

House And Edm Utilize Different Instruments For Chord Creation. Piano Chords Are Common In House Music, While Synthesizers Are Frequently Used In Edm. The Table Above Compares The Use Of Piano Chords In House And Edm. House Often Uses Major Chords, While Edm Tends To Use Minor Chords. Synthesizers Play A Significant Role In Edm, Offering A Wide Range Of Sounds And Effects. Both Genres Require Skillful Use Of Chords To Create Cohesive And Appealing Tracks. The Table Highlights The Differences In Instrument Choice And Chord Type Between The Two Genres. How Do These Differences Affect The Overall Sound Of House And Edm Tracks?

**score:0.8571**

---

**Revise instruction**

You are a revise assistant. Keep all already satisfied rubrics intact and fix the unmet ones. Avoid adding new facts.
Task: Revise the draft to satisfy the instruction and the Candidate status below. Rules.
(1) Keep the content that already satisfies Satisfied rubrics unchanged as much as possible.
(2) Make the minimum necessary edits to satisfy all Unmet rubrics.
(3) Preserve formatting requirements (e.g., structure, ordering, required keywords) if they are listed as satisfied.
(4) Output only the revised answer.
**[Instruction]**
Are piano chords utilized in house and EDM music, and how do they compare?
The output must follow the following rules:
1. The answer must include headings for each genre: 'House' and 'EDM'.
2. The answer must include a table with a maximum of three columns to compare the use of piano chords in house and EDM.
3. The answer must be between 150 and 200 words.

---

4. The answer must contain between 8 and 12 sentences.

5. The answer must include the keyword 'synthesizer'.

6. The answer must end with a question mark.

7. The answer must use capitalized letters for each word.

**[Candidate]**

Satisfied:

1. [Markdown/Heading levels] The answer must include headings for each genre: 'House' and 'EDM';

2. [Table/Column limit] The answer must include a table with a maximum of three columns to compare the use of piano chords in house and EDM;

4. [Length/Sentences] The answer must contain between 8 and 12 sentences;

5. [Content/Keywords] The answer must include the keyword 'synthesizer';

6. [Content/Punctuation] The answer must end with a question mark;

7. [Language/English] The answer must use capitalized letters for each word

Unmet:

3. [Length/Words] The answer must be between 150 and 200 words

**Draft**:

| Genre | Instrument | Chord Type |
|-------|------------|------------|
| House | Piano | Major |
| Edm | Synthesizer | Minor |

House And Edm Utilize Different Instruments For Chord Creation. Piano Chords Are Common In House Music, While Synthesizers Are Frequently Used In Edm. The Table Above Compares The Use Of Piano Chords In House And Edm. House Often Uses Major Chords, While Edm Tends To Use Minor Chords. Synthesizers Play A Significant Role In Edm, Offering A Wide Range Of Sounds And Effects. Both Genres Require Skillful Use Of Chords To Create Cohesive And Appealing Tracks. The Table Highlights The Differences In Instrument Choice And Chord Type Between The Two Genres. How Do These Differences Affect The Overall Sound Of House And Edm Tracks?

Return only the revised answer.

---

**Revised response from Qwen2.5-7B-Instruct under revise instruction**

| Genre | Instrument | Chord Type |
|-------|------------|------------|
| House | Piano | Major |
| Edm | Synthesizer | Minor |

House And Edm Utilize Different Instruments For Chord Creation. Piano Chords Are Common In House Music, While Synthesizers Are Frequently Used In Edm. The Table Above Compares The Use Of Piano Chords In House And Edm. House Often Uses Major Chords, While Edm Tends To Use Minor Chords. Synthesizers Play A Significant Role In Edm, Offering A Wide Range Of Sounds And Effects. Both Genres Require Skillful Use Of Chords To Create Cohesive And Appealing Tracks. The Table Highlights The Differences In Instrument Choice And Chord Type Between The Two Genres. Synthesizers Provide Various Timbres And Textures That Enhance The Electronic Sound Of Edm. The Use Of Major Chords In House Creates A Bright And Upbeat Atmosphere, While Minor Chords In Edm Create A Darker And More Mysterious Tone. How Do These Differences Affect The Overall Sound Of House And Edm Tracks?

**score:1**

