# OpenReview forum: "Experience is the Best Teacher: Motivating Effective Exploration in Reinforcement Learning for LLMs"
_ICML.cc/2026/Conference — ICML 2026 regular_

### Official Review · Reviewer_EY2q · 2026-03-09

**Soundness:** 3
**Presentation:** 3
**Significance:** 2
**Originality:** 2
**Overall Recommendation:** 4
**Confidence:** 4

**Summary:**

This paper presents HeRL, a framework that improves reinforcement learning exploration for large language models by using rubric-based feedback. HeRL identifies near-success trajectories, diagnoses their unmet criteria, and prompts the model to generate a rewritten response. This revision, paired with its rubric feedback, forms a "hindsight experience" to train the policy, avoiding inefficient blind trial-and-error. A bonus reward further incentivizes responses demonstrating high improvement potential. Experiments across instruction-following, writing, and medical reasoning tasks show HeRL consistently outperforms baselines like SFT, DPO, and standard RLVR.

**Compliance With Llm Reviewing Policy:**

Affirmed.

**Final Justification:**

The authors have addressed my main concerns, therefore, I raised my score from 3 to 4.

**Key Questions For Authors:**

See above

**Limitations:**

yes

**Strengths And Weaknesses:**

* **Soundness:** I feel that the paper is generally sound. The method is simple and intuitive, and the experiments show it works well across different domains.
  * The experiments look solid. The authors tested three different models on several benchmarks, including in-domain and out-of-distribution tasks. So I think the experiments meet the basic bar for being sufficient.
  * However, I actually think a few more baselines would make the paper much stronger:
    * First, what if we just collect the revisions and do SFT on those trajectories?
    * Second, in Table 2, it would be great to see the RLVR baseline evaluated for OOD performance as well.
    * Third, the result on the right side of Figure 4 is not very informative. Comparing cascade sampling to simple Pass@k does not seem completely fair since Cascade gets more info about the final rubrics. A fairer test may be to give the base model the rubrics too, so both have the same information.
    * Fourth, in Table 3, it would help to ablate just the BR alone. This means training the model without its revisions, using the revisions only to give a more fine-grained reward than just correctness.

* **Presentation:** The presentation is generally good, but some parts are a bit confusing.
  * Section 2.2 is described pretty vaguely. I actually don't know exactly how "guided sampling by hindsight experience" functions when reading from the beginning to that point. The authors should either explain this setting right there, or maybe move it to the beginning of the experimental section if it takes too many words to explain early on.


* **Significance:** I think this paper tackles an important topic. It connects well to the broad community working on RL with rubrics. It offers a nice, additional way to actually utilize rubrics in training.

**Originality:** The work brings a fresh perspective to using rubrics. Instead of just using rubrics to give an outcome reward, this paper shows how they can be used to design an auxiliary task. This is a neat idea.

---

Overall, I tend to give it a weak reject right now because of the experimental concerns I raised above. But I am willing to raise my score if the authors can add (some of) these experiments.

---

> ### Author Rebuttal · Authors · 2026-03-31
>
> We greatly appreciate the reviewer for your insightful comments and positive support.
>
> **[Q1] SFT on the revision trajectories.**
>
> Table R1. Comparison with SFT and SFT w/ Revision.
> |Models| Setting      | IFEval | IFBench | MulDimIF | WritingBench | LLMEval-Med | HealthBench |
> |:-----: | ------------ | -----: | ------: | -------: | -----------: | ----------: | ----------: |
> |Qwen2.5-7B-Instruct | SFT|75.6 |27.9 |     67.8 |         51.5 |        34.8 |        27.2 |
> ||SFT w/ Revision|70.3|27.1|62.4|53.2|55.9|27.6|
> ||HeRL|82.4|39.7|83.4|59.1|65.0|34.3|
>
> We tested this alternative by collecting revision trajectories on the training set and using them for supervised fine-tuning (“SFT w/ Revision”). The results show that this does not recover the benefit of HeRL: SFT Revision is not only much weaker than HeRL, but also underperforms standard SFT on IFEval, IFBench, and MulDimIF. This suggests that the gain of HeRL is not from simply adding rewritten responses as extra SFT data. We believe the reason is that revision trajectories are biased and heterogeneous supervision generated from failed attempts; direct SFT treats them as uniformly correct targets, which can hurt general instruction-following. In contrast, HeRL uses them within reward-guided RL training rather than pure imitation. We do observe relatively larger gains on medically oriented evaluation, especially LLMEval-Med, likely because medical rubrics are more fine-grained and thus provide richer supervision.
>
> **[Q2] RLVR results in ood benchmark.**
>
> Thank you for pointing this out. We have added the OOD evaluation results, and we will also revise the manuscript accordingly.
> Table R2. Performance of HeRL and RLVR on out-of-domain benchmarks.
>
> | Model                  | MATH-500 | GPQA | MMLU-Pro |
> | ---------------------- | -------: | ---: | -------: |
> | Llama-3.2-3B-Instruct  |     38.6 | 26.5 |     33.9 |
> | + RLVR                 |     39.6 | 29.2 |     35.1 |
> | + HeRL (Ours)          |     36.6 | 29.2 |     34.8 |
> | Qwen2.5-7B-Instruct    |     76.0 | 34.1 |     57.3 |
> | + RLVR                 |     74.6 | 34.3 |     54.3 |
> | + HeRL (Ours)          |     77.6 | 33.9 |     55.5 |
> | Qwen3-4B-Instruct-2507 |     86.0 | 46.6 |     61.5 |
> | + RLVR                 |     88.0 | 43.0 |     61.4 |
> | + HeRL (Ours)          |     89.8 | 45.3 |     62.1 |
>
> **[Q3] RLVR cascade  sampling vs HeRL cascade sampling.**
>
> We agree that comparing cascade sampling against plain pass@k is not a fully controlled test, since cascade receives additional rubric feedback. To address the fairness concern, we further compare RLVR + Cascade and HeRL + Cascade, where both methods use the same cascade protocol and are given the same rubric feedback at each round. Under this controlled setting, HeRL still consistently outperforms RLVR. This indicates that the benefit is not merely due to providing extra rubric information, but that HeRL-trained models are better able to leverage such feedback for iterative self-improvement.
>
> Table R3. RLVR cascade  sampling vs HeRL cascade sampling
> | Model| Benchmark | Round | RLVR + Cascade | HeRL + Cascade | Gain |
> | ------------------- | --------- | ----: | -------------: | -------------: | ---: |
> | Qwen2.5-7B-Instruct|MulDimIF|1 |73.5 |83.4 | +9.9 |
> |||2|77.4|87.3|+9.9|
> |||3|79.9|87.5|+7.6|
> |||4|82.3|87.5|+5.2|
>
> |Model|Benchmark|Round|RLVR+Cascade|HeRL+Cascade|Gain |
> | ---------------------- | ----------- | ----: | -------------: | -------------: | ---: |
> | Qwen3-4B-Instruct-2507|HealthBench|1|41.7 |43.5|+1.8|
> |||2|59.5|65.1|+5.6|
> |||3|63.6|70.6|+7.0|
> |||4|68.2|72.3|+4.1|
>
>
>
> We thank the reviewer for pointing this out. We will include these additional experiments and their corresponding results in the revised manuscript.
>
>
>
> **[Q4]Addtional BR only  ablation.**
>
> Table R4. BR-only ablation.
> | Models| Setting     | IFEval | IFBench | MulDimIF | WritingBench | LLMEval-Med | HealthBench |
> | :------------------ | ----------- | -----: | ------: | -------: | -----------: | ----------: | ----------: |
> | Qwen2.5-7B-Instruct | RLVR|77.3|31.6|73.5|54.8|60.5|30.5|
> ||RLVR w/ BR-only |80.1|33.6|75.7|57.9 |62.6|28.3 |
> ||HeRL|82.4 |39.7|83.4|59.1|65.0 |34.3 |
>
> We additionally include a BR-only ablation, where we do not incorporate rubric diagnose revision trajectories into training. The results show that BR-only consistently outperforms RLVR， introducing BR provides a more informative learning signal.
>
> **[W1] Section 2.2 is described pretty vaguely.**
>
> We agree that the current description of Section 2.2 is not sufficiently clear. In particular, HeRL treats initial failed trajectories along with their unmet rubrics as hindsight experience, which serves as in-context guidance to generate next-round responses. We will revise Section 2.2 in the manuscript to state it explicitly and easier to follow from the first introduction of the method.

---

> > ### Author Rebuttal · Reviewer_EY2q · 2026-04-04
> >
> > Thank you for the clarification. Regarding Q3, I still believe it is essential to provide results for the base model in addition to the RLVR model to ensure a fair comparison. Could the authors please report this metric for the base model as well?

---

> > > ### Author Response · Authors · 2026-04-04
> > >
> > > Thanks for pointing this out. We have further added Base + Cascade as an additional control under the same setting. HeRL + Cascade remains the strongest variant across all rounds as shown in talbe RR1. These results further support that the gain is not merely from adding rubric diagnosis during inference, but from HeRL-trained models being better able to leverage such diagnosis for iterative self-improvement.
> > >
> > > Table RR1. Cascade sampling comparison on MulDimIF and HealthBench
> > > |Model|Benchmark|Round|Base + Cascade|RLVR + Cascade|HeRL + Cascade|
> > > |-|-|-:|-:|-:|-:|
> > > | Qwen2.5-7B-Instruct | MulDimIF  |1 |51.4 |73.5 |**83.4** |
> > > |||2 |58.7 |77.4 |**87.3** |
> > > |||3 |59.1 |79.9 |**87.5** |
> > > |||4 |59.3 |82.3 |**87.5** |
> > >
> > > |Model|Benchmark|Round|Base + Cascade|RLVR + Cascade|HeRL + Cascade|
> > > |-|-|-:|-:|-:|-:|
> > > | Qwen3-4B-Instruct-2507 | HealthBench |1 |42.0 |41.7 |**43.5** |
> > > |||2 |60.0 |59.5 |**65.1** |
> > > |||3 |63.3 |63.6 |**70.6** |
> > > |||4 |68.9 |68.2 |**72.3** |

---

### Official Review · Reviewer_tvKt · 2026-03-10

**Soundness:** 3
**Presentation:** 3
**Significance:** 3
**Originality:** 3
**Overall Recommendation:** 4
**Confidence:** 4

**Summary:**

This paper focuses on Reinforcement learning with Rubric rewards, where they propose HeRL (Hindsight experience guided Reinforcement Learning), by treating failed attempts with unmet rubrics as in-context guidance to explore desired responses beyond initial context while incorporating an additional reward to incentivize responses that lead to improvement. Experimental results demonstrate that HeRL outperforms standard baselines.

**Compliance With Llm Reviewing Policy:**

Affirmed.

**Final Justification:**

The rebuttal addressed my main concerns

**Key Questions For Authors:**

## Questions
- It seems only one hindsight experience is being used during RLVR training. What is the effect of increasing hindsight experience while decreasing the number of rollouts?
- What is the computational cost of Hindsight Experience compared to GRPO?

**Limitations:**

yes

**Strengths And Weaknesses:**

## Strength
- The methods seem novel, inspired by traditional RL
- Strong empirical results demonstrate the effectiveness of HeRL across a diverse range of tasks.

## Weaknesses
- Lack of baseline: While the work mentions other exploration approaches, such as structured exploration and intrinsic rewards, these methods are not included in the experimental comparisons.

---

> ### Author Rebuttal · Authors · 2026-03-31
>
> We sincerely appreciate the reviewer for finding that our paper is well organized and well motivated, with reasonable and detailed experiments.
>
> **[W1] Lack of baselines.**
>
> Thanks for your suggestion. We further expand our comparisons to training-based methods like TreeRL(structured exploration)[1], DAPO(intrinsic rewards) [2] in Table. On the shared benchmarks, HeRL consistently outperforms all compared methods.
>
> Table R1. Comparison with training-based methods
> | Models | Setting | IFEval | IFBench | MulDimIF | WritingBench |
> | :------------------ | ------- | -----: | ------: | -------: | -----------: |
> | Qwen2.5-7B-Instruct | RLVR    |   77.3 |    31.6 |     73.5 |         54.8 |
> || DAPO| 76.2 |    30.2 |     63.0 |         59.0 |
> || TreeRL|   78.5 |    34.0 |     73.0 |         57.9 |
> || HeRL|   82.4 |    39.7 |     83.4 |         59.1 |
>
>
>
>
>
> **[Q1] Number of hindsight experience.**
>
> In fact increasing to 2-3 rewrites per group will not improve results Although our method uses rubric diagnosis to guide the model to revise the original trajectory into a better rewritten one, the guided response is effectively off-policy with respect to the current policy. For this reason, we use only one hindsight experience(HE) during training to limit the off-policy deviation. Empirically, Table R2 shows that increasing HE to 2 leads to lower benchmark performance, suggesting that adding more rewritten samples does not provide additional benefit and instead introduces bias.
>
> Table R2. Comparison with different numbers of hindsight experience.
> | Model   | Setting | IFEval | IFBench | MulDimIF | WritingBench | LLMEval-Med | HealthBench |
> | ------- | ------- | ------ | ------- | -------- | ------------ | ----------- | ----------- |
> | Qwen2.5-7B-Instruct | HE=1    | 82.4   | 39.7    | 83.4     | 59.1         | 65.0        | 34.3        |
> |  | HE=2    | 81.9   | 38.5   | 82.7     | 56.0         | 61.9        | 33.2        |
>
> **[Q2] What is the computational cost of Hindsight Experience compared to GRPO?**
>
> Table R3. Computational cost comparsion between GRPO and HeRL.
> | Method | Time / Step (min) | Trajectories / Step |
> |---|---:|---:|
> | GRPO | 2.31 | 512 |
> | HeRL | 3.51 | 598 |
>
> Under the same experimental setup, the training logs show that GRPO samples 512 trajectories per step, while HeRL samples 598 trajectories per step on average. The corresponding average per-step cost on 4×A800 GPUs is 2.31 minutes for GRPO and 3.51 minutes for HeRL.
>
> ---
>
> [1]Hou et al. TreeRL: LLM Reinforcement Learning with On-Policy Tree Search. ACL 2025
>
> [2]Yao et al. Diversity-Aware Policy Optimization for Large Language Model Reasoning. NeurIPS 2025

---

> > ### Author Rebuttal · Reviewer_tvKt · 2026-04-03
> >
> > Thank you for your response. My questions have been fully resolved. I will keep my score and increase my confidence, leaning toward acceptance.

---

> > > ### Author Response · Authors · 2026-04-04
> > >
> > > We sincerely thank the reviewer for the positive assessment and recommendation of our work. We particularly appreciate the recognition that our method is novel and well-motivated, as well as the acknowledgment of the strong empirical results showing the effectiveness of HeRL across diverse tasks. We are also grateful for the constructive comments and suggestions, which have helped improve the quality and clarity of the manuscript.

---

### Official Review · Reviewer_WkCD · 2026-03-12

**Soundness:** 3
**Presentation:** 3
**Significance:** 3
**Originality:** 3
**Overall Recommendation:** 4
**Confidence:** 4

**Summary:**

The authors use rubric-based rewarding and find near-success rollouts (ones with the maximum reward) and rewrite the answer for them while conditioning on the rubric to get better answers. They take the final best answer, include that in the GRPO group, and also assign a bonus to the reward of the original answer it came from. The results show that this significantly improves pass@k.

**Compliance With Llm Reviewing Policy:**

Affirmed.

**Final Justification:**

This is a good technique to improve the performance of GRPO. It shows a good path towards techniques tailored for multi-turn LLM interactions. However, I think that while this is a great direction, there can be more effort to understand why and how that best answer improves performance, leading to a more *elegant* hypothesis or algorithm.

**Key Questions For Authors:**

- You fix hindsight_experience=1. Does increasing to 2-3 rewrites per group (with corresponding bonus rewards) improve results, or do diminishing returns kick in quickly?
- All rubric evaluations use GPT-4o mini. Have you tried weaker judges (e.g., an 8B model)? If the gains disappear with a weaker judge, it suggests the method is benefiting from distillation rather than the rubric structure itself.
- You show cascaded rewriting helps at test time (Figure 4 right). Did you try iteratively rewriting trajectories during training (rewrite -> re-score -> rewrite again)? If so, what went wrong?
- You mask the entire prefix H. Did you try computing loss on the source trajectory tokens as well (while still masking the rubric diagnosis)? Curious whether there's useful signal in also training on the original attempt within the hindsight context, or if it just introduces noise.

**Limitations:**

yes

**Strengths And Weaknesses:**

Strengths:
- Insightful ablations that show assigning credit to the original answer is impactful.
- Improving pass@k is significant for real-world applications.
- Simple explanations. The trivial proposition 3.1 was unnecessary, though.

Weaknesses:
- Not providing code. This raises concerns about the reproducibility, especially since the table of results doesn't include std values.
- The cascade results lack reward-based in-context learning baselines such as [1] or [2].

---

[1] Song et al. Reward Is Enough: LLMs Are In-Context Reinforcement Learners. 2025

[2] Yao et al. Tree of Thoughts: Deliberate Problem Solving with Large Language Models. 2023

---

> ### Author Rebuttal · Authors · 2026-03-31
>
> We sincerely appreciate the reviewer for finding that our paper is well organized and well motivated, with reasonable and detailed experiments.
>
> **[W1] Proposition 3.1 was unnecessary.**
>
> Thank you for pointing this out. Proposition 3.1 was included mainly to formalize the proposed method.  We agree that this part can be made more concise, and we will simplify and refine its presentation in the revised manuscript.
>
> **[W2] Concern about the reproducibility.**
>
> The code has been open-sourced and is available at: **https://anonymous.4open.science/r/HeRL-DCBB/**. To address the concern about reproducibility, we set the temperature to 0.7 and repeat the evaluation five times to compute the std, as shown in Table R1 **https://anonymous.4open.science/r/HeRL-DCBB/captions/tableR1.pdf**.
>
> **[W3] Baseline method are not comprehensive.**
>
> Table R2 Comparsion with testtime in-context learning baselines.
>  Models|Setting|HealthBench-500|
> |-|-|-:|
> |Qwen3-4B-Instruct-2507 + HeRL|Pass@4|55.00|
> ||ToT|44.85|
> ||ICRL|67.60|
> ||Cascade |72.30|
>
> |Models|Setting|MulDimIF|
> |-|-|-:|
> |Qwen2.5-7B-Instruct + HeRL|Pass@4|86.00|
> ||ToT|84.40|
> ||ICRL|86.30|
> ||Cascade|87.50|
>
> We additionally evaluate **ICRL**[1] and **ToT**[2] on our HeRL model. As shown in the table R2, **HeRL with cascade sampling consistently outperforms both ToT and ICRL**, indicating that the self-improvement ability learned by HeRL can be effectively activated during inference.
>
> Table R3. Comparison of different setups.
> |Model|Setting|IFEval|IFBench|MulDimIF|WritingBench|LLMEval-Med|HealthBench|
> |-|-|-:|-:|-:|-:|-:|-:|
> |Qwen2.5-7B-Instruct|HeRL|82.4|39.7|83.4|59.1|65.0|34.3|
> ||RLVR|77.3|31.6|73.5|54.8|60.5|30.5|
> ||HE=2|81.9|38.5|82.7|56.0|61.9|33.2|
> ||SFT|75.6|27.9|67.8|51.5|34.8|27.2|
> ||SFT w/ Revision|70.3|27.1|52.4|53.2|55.9|27.6|
> ||HeRL w/ Source Trajectory Loss|73.9|29.9|70.2|57.8|52.4|30.4|
>
> **[Q1] Does increasing to 2-3 rewrites per group (with corresponding bonus rewards) improve results, or do diminishing returns kick in quickly?**
>
> In fact increasing to 2-3 rewrites per group will not improve results. Although our method uses rubric diagnosis to guide the model to revise the original trajectory into a better rewritten one, the guided response is effectively off-policy with respect to the current policy. For this reason, we use only one hindsight experience(HE) during training to limit the **off-policy deviation**. Empirically, Table R3 line **HeRL and HE=2** shows that increasing HE to 2 leads to lower benchmark performance, suggesting that adding more rewritten samples does not provide additional benefit and instead introduces bias.
>
> **[Q2] Whether HeRL benefits from distillation.**
>
> We would like to note that the loss on rubric diagnosis tokens is masked, so that  the model is not trained to imitate the diagnosis itself. Second, using a weaker judge (Qwen3-8B) degrades both RLVR and HeRL, suggesting that weaker criterion-level judgments introduce noisier reward signals as shown in table R4. Third, directly using collected revision trajectories for supervised fine-tuning still performs worse than HeRL, as shown in table R3 line **HeRL , SFT , SFT w/ Revision**. Together, these results suggest that HeRL’s gain does not come from distillation.
>
> Table R4. Comparison under Different Judge Models
> |Model|Method|Judge|HealthBench|LLMEval-Med|
> |-|-|-|-:|-:|
> |Qwen2.5-7B-Instruct|RLVR|GPT-4o-mini|30.5|60.5|
> ||HeRL|GPT-4o-mini|34.3|65.0|
> ||RLVR|Qwen3-8B|27.5|53.5|
> ||HeRL|Qwen3-8B|30.3|58.4|
>
> **[Q3] Rewritting mutiple rounds.**
>
> We agree that two rounds of rewriting can further improve training, but we use one rewrite round. Empirically, the second rewrite yields only limited marginal gain, bringing about **10% more HE samples**. At the same time, it increases the average per-step cost from **3.33 to 4.36 GPU-minutes** and raises the rewrite rollout cost to 1.82× that of one-round rewriting. We therefore adopt one-round rewriting in training.
>
> **[Q4] Is there useful training signal in the source trajectory tokens?**
>
> As shown in the table R3 line **HeRL, RLVR, HeRL w/ Source Trajectory Loss**, computing loss on the source trajectory tokens does not help; in fact, its performance is even **below RLVR**, suggesting that the source trajectory introduces substantial noise rather than useful supervision in this hindsight context. Our interpretation is that the original trajetory mainly serves as guidance for identifying what failed and guiding the revision, but forcing the model to imitate those tokens also reinforces suboptimal generations.
>
> ---
> [1] Song et al. Reward Is Enough: LLMs Are In-Context Reinforcement Learners. NeurIPS 2025
>
> [2] Yao et al. Tree of Thoughts: Deliberate Problem Solving with Large Language Models. NeurIPS 2023

---

> > ### Author Rebuttal · Reviewer_WkCD · 2026-04-04
> >
> > I thank the authors for their detailed experiments. I've increased my confidence score.

---

> > > ### Author Response · Authors · 2026-04-04
> > >
> > > We appreciate the reviewer’s encouraging evaluation of our paper. We particularly appreciate the reviewer’s recognition of our ablation experiments and the overall clarity of the presentation. We are grateful for the reviewer’s comments, which have helped improve the manuscript and clarify several points.

---

### Official Review · Reviewer_3LSZ · 2026-03-15

**Soundness:** 3
**Presentation:** 2
**Significance:** 2
**Originality:** 2
**Overall Recommendation:** 4
**Confidence:** 3

**Summary:**

This paper presents an in context learning approach towards exploration leveraging rubrics and textual feedback. The paper also uses a bonus rewards incentivizing responses with the greatest chance of improvements. The paper uses three classes of benchmarks to evaluate the efficacy of their proposed framework.

**Compliance With Llm Reviewing Policy:**

Affirmed.

**Key Questions For Authors:**

1. Can the authors clarify how the quality of a generation satisfying a rubric is verified? How robust is the algorithm to mis-specification of these scores (which in turn influences the rewards)? Are there any ablations that specifically attempt to tease apart the robustness of reward estimation?
2. Can the authors provide some quantitative treatment as to whether their proposed strategy allows the LLMs to get more diverse generations that help cover the set of rubrics (this is somewhat like pass@k for reasoning, but measured wrt the rubrics in question)?

**Limitations:**

yes

**Strengths And Weaknesses:**

Presentation: Generally, I found the experimental section to be harder to follow given the switches between different setups etc, but the rest of the paper felt more easier to understand.

Significance: This paper considers an important problem but in terms of techniques etc, the significance is mixed. Furthermore, as the authors themselves acknowledge, the applicability of these techniques are conditioned on having rubrics and ways to evaluate generations against these rubrics, both of which limit the scope.

Originality: The paper's use of rubrics plus textual feedback to guide exploration bears relationship to several works that utilize textual feedback for guiding LLM generations, perhaps not exactly for the purposes of exploration.

Soundness: Relatively sound in terms of experimental results (questions below).

---

> ### Author Rebuttal · Authors · 2026-03-31
>
> We greatly appreciate the reviewer for your insightful comments and positive support.
>
> **[W1]  Hard to follow the switches between different experiment setups.**
>
> At the beginning of Section 4, we outline the overall logic of our experiments. **Section 4.1** evaluates the performance of HeRL against baselines on the main benchmarks. **Section 4.2** examines whether HeRL preserves out-of-distribution generalization ability. **Section 4.3** studies whether the self-improvement behavior learned from hindsight experience can also emerge at test time; specifically, since HeRL is trained on hindsight experience samples, we evaluate this by using multi-round rewriting at inference time. **Section 4.4** presents ablation studies to verify that both hindsight experience and the bonus reward contribute to the final gains. **Section 4.5** further analyzes training dynamics, including entropy, to show that our method indeed encourages exploration during training. In the revision, we will further improve the opening of Section 4, as well as the presentation of the cascade setting, to make the experimental setup and the goal of each subsection easier to follow.
>
> **[W2] Rubrics limit the applicability.**
>
> Our method is decoupled from rubrics. It only requires criterion-level diagnosis indicating whether each rubric criterion is satisfied, and can therefore fully leverage the rich semantic information encoded in the rubric. This makes the method more general than approaches tied to a specific rubric design. Moreover we use rubric diagnosis as structured diagnosis to guide rewriting and self-improvement. HeRL is also naturally compatible with future advances such as self-evolving rubrics[1] or automatically generated rubric diagnosis, suggesting that the framework is inherently scalable.
>
> **[W3] Not exactly for the purposes of exploration.**
>
> Our method is specifically motivated by improving **exploration during generation** under guidance from criterion-level satisfaction diagnosis, so as to produce more diverse samples and enrich the training signal. Moreover, this ability to use rubric diagnosis for **self-improvement** can be internalized by the model itself, as shown in Figure 4 in this paper. We also provide evidence from training dynamics(Figure 5 in this paper): compared with RLVR, HeRL yields significantly higher entropy, which supports that our method indeed improves the model’s exploration ability.
>
> **[Q1] How the rubrics is verified and its robustness.**
>
> The quality verification follows the rubric decomposition in Eq. 5. Each rubric typically contains multiple criteria, and for each criterion we use an LLM-as-a-judge or rule-based method to determine whether it is satisfied, corresponding to the binary indicator in the equation. The final reward is then computed from the weighted sum of criterion-level satisfaction signals. In this sense, the score depends on the accuracy of the criterion judgments, rather than on a single holistic score, which makes the reward more structured and interpretable.
>
> Regarding robustness to score mis-specification, the main source of noise comes from errors in judging whether each criterion is satisfied. In our case, this noise appears limited. On training samples generated by Qwen2.5-7B-Instruct, when we scored each sample five times, GPT-4o-mini gave a reward mean of **0.617±0.012**, DeepSeek-V3 gave **0.596±0.020**, and Qwen3-8B gave **0.505±0.037**. This suggests that Qwen3-8B exhibits a bias in rubric-based judgment.
>
> This trend is also reflected in downstream training results in Table R1. Replacing GPT-4o-mini with Qwen3-8B leads to lower performance for both RLVR and HeRL, indicating that training quality is indeed affected by reward accuracy. However, under the same judge model, HeRL consistently outperforms RLVR. This suggests that while absolute performance depends on judge quality, the advantage of HeRL over RLVR is robust across different judges.
>
> Table R1. Comparison under Different Judge Models
> |Model|Method|Judge|HealthBench|LLMEval-Med|
> |-|-|-|-:|-:|
> |Qwen2.5|RLVR|GPT-4o-mini|30.5|60.5|
> ||HeRL|GPT-4o-mini|34.3|65.0|
> ||RLVR|Qwen3-8B|27.5|53.5|
> ||HeRL|Qwen3-8B|30.3|58.4|
>
>
>
> **[Q2] Generation diversity on rubrics.**
>
> Under the reward definition in Eq. 5, a higher score suggests that the model satisfies a broader set of criteria, and therefore reflects stronger criterion-level coverage. From this perspective, the reward is related not only to quality but also to the diversity of covered criteria. This is consistent with Table R2, where HeRL consistently achieves higher pass@k than both RLVR and the base model, indicating stronger exploration and more diverse successful generations.
>
>
> Table R2. Qwen2.5-7B-Instruct pass@k performance.
> |k|Base|RLVR|HeRL|
> |-:|-:|-:|-:|
> |1|26.5|33.6|39.7|
> |8|41.1|46.5|53.0|
> |64|50.6|54.0|59.5|
> |256|58.8|59.5|61.5|
>
> ---
> [1] Rezaei et al. Online Rubrics Elicitation from Pairwise Comparisons. 2026

---

> > ### Author Rebuttal · Reviewer_3LSZ · 2026-04-04
> >
> > I believe the authors addressed the concerns to the extent possible, but the use of the techniques being tied to alignment problems that have established rubrics associated with them already limit (at least to some extent) the applicability of the approach itself. I will retain my score as is, and thank the authors for their response in addressing the other questions.

---

> > > ### Author Response · Authors · 2026-04-04
> > >
> > > We appreciate the reviewer’s thoughtful comment and positive support for our work. We note that rubric-based supervision has already been explored in multiple domains, including instruction following (ComplexBench[1]), scientific and medical reasoning (RaR[2]), legal reasoning (PLawBench[3]), deep research across diverse real-world domains (ResearchRubrics[4]), and general chatbot evaluation (CHARM-100[5]). In addition, RubricHub[6] provides a large scale (~110k), multi-domain rubric dataset. We therefore view the current scarcity of rubric-based resources as a broader limitation of the current community, rather than a restriction introduced by HeRL itself. Looking forward, we believe the continued development of rubric datasets and evaluators will make this form of supervision increasingly accessible.
> > >
> > > ---
> > > [1] Wen et al. Benchmarking Complex Instruction-Following with Multiple Constraints Composition 2024
> > >
> > > [2] Gunjal et al. Rubrics as Rewards: Reinforcement Learning Beyond Verifiable Domains. 2025
> > >
> > > [3] Shi et al. PLAWBENCH: ARubric-Based Benchmark for Evaluating LLMs in Real-World Legal Practice. 2026
> > >
> > > [4] Sharma et al. ResearchRubrics: A Benchmark of Prompts and Rubrics For Evaluating Deep Research Agents. 2025
> > >
> > > [5] Rao et al. Autorubric: A Unified Framework for Rubric-Based LLM Evaluation. 2026
> > >
> > > [6] Li et al. RubricHub: A Comprehensive and Highly Discriminative Rubric Dataset via Automated Coarse-to-Fine Generation. 2026

---

### Decision · Program_Chairs · 2026-04-30

**Decision:**

Accept (regular)

**Comment:**

This paper presents HeRL (Hindsight experience guided Reinforcement Learning), a framework that improves reinforcement learning exploration for large language models with rubric rewards. HeRL treats failed attempts with unmet rubrics as in-context guidance to explore desired responses beyond initial context while incorporating an additional reward to incentivize responses that lead to improvement.  Experiments across instruction-following, writing, and medical reasoning tasks show HeRL consistently outperforms baselines like SFT, DPO, and standard RLVR.

The reviewers all appreciated the problem being important and the algorithm being simple and intuitive. The reviewers believe using rubrics plus textual feedback is a good technique to improve the performance of GRPO.  The concerns were mostly regarding lack of experiments and the rebuttal provided results on new baselines, new jduge models, ablation on different hyperparameters, etc. One reviewer increased the score after rebuttal, hence all reviewers are positive. I agree with the reviewers' consensus and recommend acceptance.